# Where are we with calibration under dataset shift in image classification?

**Mélanie Roschewitz**                                  *m.roschewitz21@imperial.ac.uk*
*Imperial College London*

**Raghav Mehta**                                        *raghav.mehta@imperial.ac.uk*
*Imperial College London*

**Fabio De Sousa Ribeiro**                              *f.de-sousa-ribeiro@imperial.ac.uk*
*Imperial College London*

**Ben Glocker**                                         *b.glocker@imperial.ac.uk*
*Imperial College London*

**Reviewed on OpenReview:** *https://openreview.net/forum?id=1NYKXlRU2H*

## Abstract

We conduct an extensive study on the state of calibration under real-world dataset shift for image classification. Our work provides important insights on the choice of post-hoc and in-training calibration techniques, and yields practical guidelines for all practitioners interested in robust calibration under shift. We compare various post-hoc calibration methods, and their interactions with common in-training calibration strategies (e.g., label smoothing), across a wide range of natural shifts, on eight different classification tasks across several imaging domains[1]. We find that: (i) simultaneously applying entropy regularisation and label smoothing yield the best calibrated raw probabilities under dataset shift, (ii) post-hoc calibrators exposed to a small amount of semantic out-of-distribution data (unrelated to the task) are most robust under shift, (iii) recent calibration methods specifically aimed at increasing calibration under shifts do not necessarily offer significant improvements over simpler post-hoc calibration methods, (iv) improving calibration under shifts often comes at the cost of worsening in-distribution calibration. Importantly, these findings hold for randomly initialised classifiers, as well as for those finetuned from foundation models, the latter being consistently better calibrated compared to models trained from scratch. Finally, we conduct an in-depth analysis of ensembling effects, finding that (i) applying calibration prior to ensembling (instead of after) is more effective for calibration under shifts, (ii) for ensembles, OOD exposure deteriorates the ID-shifted calibration trade-off, (iii) ensembling remains one of the most effective methods to improve calibration robustness and, combined with finetuning from foundation models, yields best calibration results overall.

## 1 Introduction

Reliable uncertainty estimation is paramount to the development of trustworthy machine learning models. Models should accurately estimate their level of uncertainty, without being over- or under-confident in their predictions. Model calibration precisely measures how well predicted confidence scores reflect the likelihood of model predictions to be correct (Murphy, 2023; Guo et al., 2017). Calibrated probabilities are more interpretable and enable reliable performance estimation post-deployment. Improving calibration has been

---

[1]Our code is publicly available at https://github.com/biomedia-mira/calibration_under_shifts

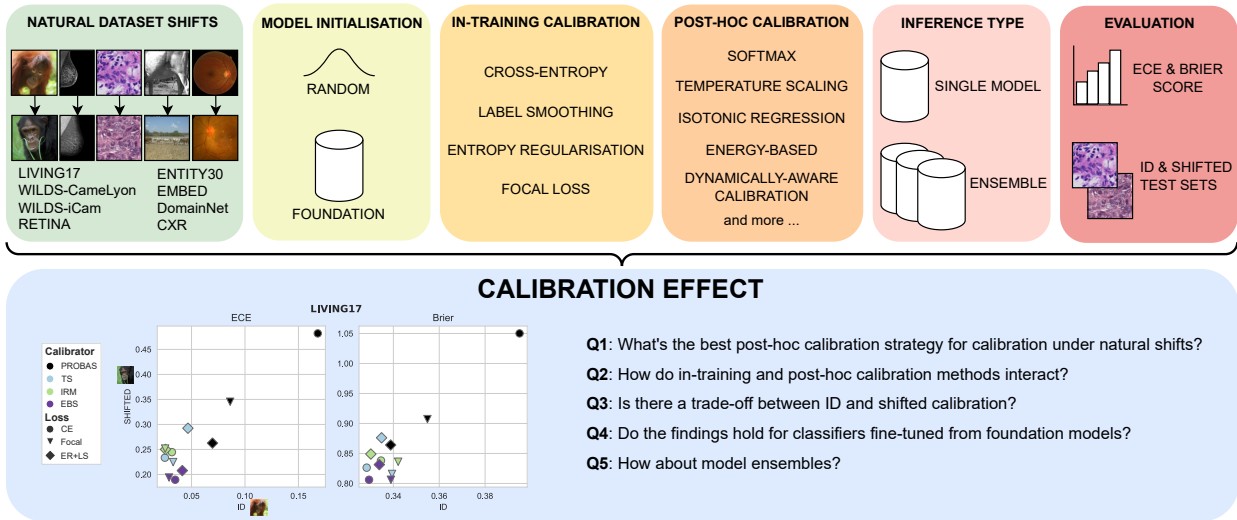

Figure 1: **Overview of our calibration study**.

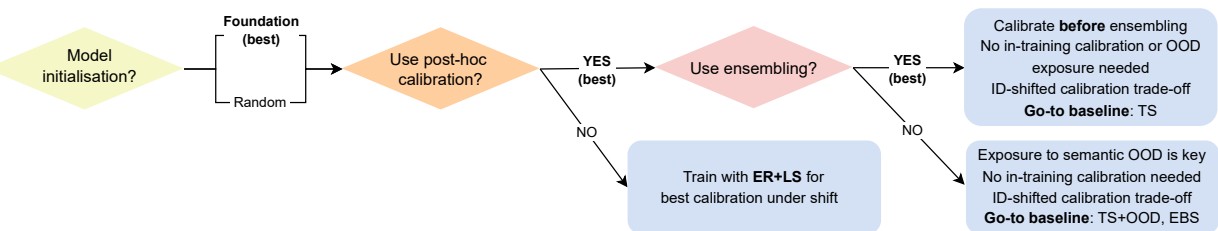

Figure 2: **Practical guidelines for improved calibration under dataset shifts**.

the topic of many papers over the last few years, across various tasks (Guo et al., 2017; Szegedy et al., 2016; Müller et al., 2019; Mukhoti et al., 2020; Zadrozny & Elkan, 2002; Gupta et al., 2021; Zhang et al., 2020; Tomani et al., 2023; 2021; Seligmann et al., 2023; Kim & Kwon, 2024).

Dataset shift is an eminent issue for model calibration (Ovadia et al., 2019). Yet, for long, most of the field has concentrated on evaluating new calibration methods in in-distribution (ID) settings, where calibration and test sets come from the same distribution (Guo et al., 2017; Zhang et al., 2020; Pereyra et al., 2017; Mukhoti et al., 2020; Meister et al., 2020). While calibration under shifts is still largely under-studied, recently, an increasing number of studies specifically focus on this issue (Tomani et al., 2021; 2023; Kim & Kwon, 2024; Yuksekgonul et al., 2023). However, most of those exclusively rely on limited synthetic corruption datasets for evaluation, i.e. CIFAR-C, ImageNet-C (Hendrycks & Dietterich, 2018; Krizhevsky, 2009; Deng et al., 2009). Unfortunately, it is known that observations made on synthetic distribution shifts may not always hold on more realistic shifts (Taori et al., 2020; Koh et al., 2021). Hence, this paper aims to better understand existing calibration methods and identify which specific factors contribute to improved calibration under *realistic* dataset shifts, across various imaging domains.

A large portion of the calibration literature has focused on developing post-hoc methods, allowing to correct model under/over-confidence after training. These, for example, include temperature scaling (Guo et al., 2017) or isotonic regression (Zadrozny & Elkan, 2002; Zhang et al., 2020). Others have instead proposed to address miscalibration directly during training, developing specific loss functions and training strategies (Mukhoti et al., 2020; Müller et al., 2019; Pereyra et al., 2017). Prominent examples include label smoothing, entropy regularisation, and focal loss, which are now widely used and part of the standard implementation of most deep learning frameworks. However, there is a need for a more in-depth look at their behaviour under natural dataset shifts. Moreover, some studies suggest that such in-training calibration methods may suffer from 'negative calibration' i.e., their calibration worsens when combined with posthoc

calibration methods (Wang et al., 2021; Zhang et al., 2022). This prompts us to not only study in-training and post-hoc calibration methods in isolation, but also to carefully examine their interaction across a wide range of shifts.

Meanwhile, foundation models have emerged as a promising avenue to improve model robustness (Oquab et al., 2023; Zhang et al., 2025; Zhou et al., 2023; Shi et al., 2024; Mehta et al., 2022). Hence, we also investigate the calibration of classifiers finetuned from these large foundation models. Specifically, we aim to understand whether: (i) the findings for randomly initialised classifiers generalise to classifiers finetuned from foundation models, (ii) calibration under dataset shifts is indeed improved compared to models trained from scratch.

Finally, perhaps the most common approach to increase robustness under shifts, when the computational budget allows, is ensembling (Lakshminarayanan et al., 2017). Many studies showed that ensembling can significantly help with model robustness and calibration (Seligmann et al., 2023; Kumar et al., 2022). However, there is also some contradictory evidence suggesting that deep ensembles are not necessarily better calibrated than individual ensemble members (Rahaman & Thiery, 2021; Kumar et al., 2022). Hence, we conclude our study by analysing ensemble calibration, comparing the effect of various calibration methods, and in particular, the impact of applying calibration before or after ensembling.

To summarise, as illustrated in Fig. 1, we comprehensively study existing in-training and post-hoc calibration methods, specifically investigating their impact on calibration under dataset shifts. *Our goal is not to propose a new method, but rather to find practical guidelines on which methods to choose for obtaining well-calibrated probability under shifts.* Our study covers eight datasets and tasks, a wide range of imaging modalities from satellite images to mammography and histopathology images. We focus on realistic distribution shifts, from geographical location shifts, medical device shifts, sub-population shifts, experimental protocols shifts, and more. We study the effect of five different in-training calibration methods and their interactions with ten post-hoc calibration methods. We evaluate models trained from random initialisation as well as finetuned from foundation models, across various CNN- and transformer-based architectures, training 420 models in total. Finally, we analyse the effect of model ensembling on calibration under shifts, compare various ensemble calibration strategies and study interactions between ensemble calibration and calibration of individual ensemble members.

The breadth of our analysis enables us to derive clear practical guidelines for practitioners interested in building classification systems with robust calibration under real-world shift, which we summarise in Fig. 2 and detail below, we find that:

(i) Classifiers fine-tuned from foundation models offer significantly better calibration (shifted and ID) compared to classifiers trained from scratch, regardless of the calibration strategy (§4.2.2);

(ii) In-training calibration is not needed when post-hoc calibration is used (§4.1.2);

(iii) Ensembling improves both ID and shifted calibration regardless of the calibration strategy (§4.3.1);

(iv) Without ensembling, the most effective strategy to improve calibration under shifts consists of adding small amounts of OOD data (unrelated to the task) to the calibration set, regardless of the choice of post-hoc calibrator (§4.1.1);

(v) For model ensembles, the best strategy to obtain robust calibration under shifts consists of applying post-hoc calibration *prior* to ensembling, and *without* OOD exposure (§4.3.2);

(vi) Improving calibration under shifts often comes at the cost of ID calibration (§4.1.3, §4.3.3).

We make all of our code available to the research community to facilitate further developments in this space.

## 2 Related work

Model calibration has been a prolific area of research. Proposed approaches notably include post-hoc calibration methods (Guo et al., 2017; Kim & Kwon, 2024; Zhang et al., 2020), alterations to loss functions (Szegedy

et al., 2016; Pereyra et al., 2017; Mukhoti et al., 2020), model ensembling (Lakshminarayanan et al., 2017) and Bayesian Neural Networks (BNN) (Detommaso et al., 2022; Maddox et al., 2019). In this study, we focus on popular calibration methods, which can be applied within standard training pipelines and for any model architecture. In particular, we consider the comparison of various BNN training strategies for calibration robustness out-of-scope. However, the reader interested in such a comparison is invited to consult Seligmann et al. (2023).

## 2.1 Model calibration metrics

We consider a classification task with $C$ classes. Let $\hat{\mathbf{p}} \in [0, 1]^C$ denote the model predicted probabilities with $\arg\max_i \hat{p}_i$ the predicted class and $y$ the associated label. A model is said to be well calibrated if the probability associated with the predicted class (a.k.a. confidence score) matches the true probability of being correctly classified (Guo et al., 2017). Several metrics have been proposed to evaluate the calibration of a given model. The most popular one is the Expected Calibration Error or ECE (Guo et al., 2017), approximating the expected difference between predicted confidence and prediction accuracy (see Appendix A.2). Despite its popularity, a major limitation of ECE (and its variants like AdaECE, SCE (Nixon et al., 2019)) resides in the necessity of introducing a binning scheme, which may influence the calibration measure and make the metric sensitive to small test set sizes (Nixon et al., 2019; Gruber & Buettner, 2022). An alternative metric is, for example, the Brier Score (Brier, 1950), which does not require any binning, is a proper score, and measures total calibration (as opposed to only top-level calibration like ECE). This score computes the average mean squared error between probabilities and one-hot labels (details in Appendix A.2).

## 2.2 In-training model calibration techniques

Several works have proposed specific loss functions and regularisation schemes tailored to improve the calibration of model outputs during training, without requiring any architectural changes. These include: label smoothing (LS) (Szegedy et al., 2016; Müller et al., 2019), entropy regularisation (ER) (Pereyra et al., 2017; Meister et al., 2020), and focal loss (Mukhoti et al., 2020). In LS, labels are softened to explicitly discourage the network from being too confident in its predictions, i.e. $\tilde{\mathbf{y}} = (1 - \lambda)\mathbf{y} + \lambda\frac{1}{C}$, with $\mathbf{y}$ the one-hot encoded labels, and $\lambda$ the hyperparameter controlling the amount of smoothing (typically ranging in $[0.01, 0.1]$). ER adds the negative entropy of the predictions to the cross-entropy loss, where the strength of the regularisation is controlled by a multiplicative constant $\alpha$ (with $\alpha = 0.1$ working well across tasks (Pereyra et al., 2017)). Another popular approach is the focal loss (Lin et al., 2020), defined as $\sum_{k=1}^{C} y_k(1 - \hat{p}_k)^\gamma \log(\hat{p}_k)$, where $\gamma$ is an hyperparameter. Mukhoti et al. (2020) show that focal loss can improve model calibration, in particular when using $\gamma = 5$ if $\hat{p}_k \in [0, 0.2)$ else $\gamma = 3$. Note that other variants of these losses have been subsequently proposed (Meister et al., 2020; Liu et al., 2022); however, here, we focus on LS, ER, and focal losses as these remain the most popular for image classification (Gao et al., 2020; Wei et al., 2022; Wang et al., 2022; Carse et al., 2022; Fuentes et al., 2023).

## 2.3 Post-hoc model calibration techniques

Modifying the loss function to improve calibration is not always a practical solution, as one may not always have full control over the training process, and loss function alterations may have inadvertent side effects, such as performance reduction. Hence, many have focused on developing *post-hoc* recalibration techniques to address mis-calibration after training. This paradigm has the advantage of being independent of the model architecture or the choice of the loss function during training and is widely used in practice. One example of such post-hoc calibration method is temperature scaling (TS) (Guo et al., 2017) (or Platt-scaling), where the logits $\mathbf{z}$ are scaled by a constant $T$, where $T$ is tuned on the validation set such that the negative likelihood is minimised. Calibrated probabilities are then obtained with $\hat{\mathbf{p}} = \text{softmax}(\mathbf{z}/T)$. Adjusting the temperature controls the 'peakiness' of the distribution, allowing to correct for model over/under-confidence. Another approach consists of using isotonic regression (IRM) (Zadrozny & Elkan, 2002) between the predicted probabilities and the expected accuracy (non-parametric). Gupta et al. (2021) have alternatively proposed to use splines for calibrating outputs (SPL). The synergies between various parametric and non-parametric post-hoc methods have been explored by Zhang et al. (2020), who notably proposed ETS, an ensemble formulation

of TS to allow for increased expressivity. They also proposed IROVaTS, combining IROVa (IRM in a one-versus-all setup for a multiclass problem) with ETS. Recently, Kim & Kwon (2024) proposed energy-based calibration (EBS), utilising the energy of the logits, i.e., $\mathcal{F}(\mathbf{z}) = -\text{logsumexp}(\mathbf{z})$ to derive a sample-wise temperature. Importantly, this method utilises both an ID validation set and an out-of-distribution (OOD) set, semantically unrelated to the task, to improve calibration on shifted data (contrary to previous methods, which do not explicitly require access to external OOD data for calibration). More details on EBS can be found in Appendix A.5.1. Importantly, TS, ETS, IRM, and EBS are all accuracy-preserving, i.e., the final class prediction is unchanged after model calibration.

## 2.4 Calibration and the problem of dataset shift

An increasing number of studies have recently focused on increasing the robustness of model calibration under dataset shift. An early study from Tomani et al. (2021) proposed to apply random augmentations to images in the validation set used to fit post-calibrators, for better results under shifts. However, the generalisation of such calibrators is bounded by the ability of these random augmentations to faithfully capture shifts encountered at test-time. Yu et al. (2022) have, on the other hand, proposed to train a domain-to-temperature regressor. However, this requires access to multiple domains at validation time and for them to capture test-time shifts reliably – a highly impractical assumption. Instead Tomani et al. (2023) proposed Density-Aware-Calibration (DAC), leveraging nearest neighbours distances in the embedding space (at different layers throughout the network) to modulate the temperature for each sample, without requiring access to multiple domains at calibration time; this idea of measuring the sample's 'atypicality' to calibrate the network is also adopted in concurrent work from (Yuksekgonul et al., 2023). Importantly, DAC is a plug-in method that can be used in conjunction with any other post-hoc calibration method. Finally, Kim & Kwon (2024) demonstrated state-of-the-art results with their EBS calibrator against a wide range of post-hoc calibrators on CIFAR10-C and ImageNet-C (Hendrycks & Dietterich, 2018), with best results obtained when EBS is used in conjunction with DAC. Yet, more analyses are needed to determine the robustness of these calibration methods on a broader set of tasks and, crucially, on real-world shifts beyond synthetic image corruptions.

For the convenience of the reader, a comparative overview of prior work on calibration under shifts in image classification and our study is presented in Table A.2.

## 2.5 Model ensembling and calibration

Another popular approach to improve calibration is model ensembling (Lakshminarayanan et al., 2017). Kumar et al. (2022) showed that ensembling diverse calibrated models yields more robust models and better calibration across various tasks. Similarly, a recent study from Seligmann et al. (2023) on the calibration of Bayesian neural networks showed that ensembling was more efficient than any single Bayesian calibration method in terms of robustness under shifts. However, others report contradictory evidence as well. Wu & Gales (2021) provided theoretical insights on why calibration of individual ensemble members is not sufficient for their aggregate predictions to be well calibrated, and Rahaman & Thiery (2021) showed that deep ensembles are not necessarily better calibrated than individual ensemble members. These contradictory findings prompt us to include ensembling effects in our study.

# 3 Experimental setup

## 3.1 Datasets and studied shifts

We analyse calibration robustness across different image classification tasks and real-world natural distribution shifts. First, we investigate robustness against geographic and sub-population shifts, in natural image classification using the **Living17** and **Entity30** datasets (Santurkar et al., 2020), as well as **Wilds-iCam** (Koh et al., 2021). Next, we analyse realistic shifts specific to medical imaging, a high-stakes domain where robustness and calibration are of particular importance. We study calibration robustness against: (i) scanner changes in breast density assessment models using the **EMBED** (Jeong et al., 2023) mammography dataset, (ii) scanner, population and prevalence changes in chest X-ray classification (No Finding /

| Dataset | Type of shift(s) | Task | N classes |
|---------|------------------|------|-----------|
| Living17 | Population | Balanced | 17 |
| Entity30 | Population | Balanced | 30 |
| CameLyon | Staining | Balanced | 2 |
| CXR | Scanner, Prevalence, Population | Imbalanced | 2 |
| RETINA | Scanner, Prevalence, Population | Imbalanced | 5 |
| EMBED | Scanner | Imbalanced | 4 |
| iCam | Geographic, Prevalence | Imbalanced | 182 |
| DomainNet | Image type, Prevalence | Imbalanced | 345 |

Table 1: Datasets and shifts used in this study.

Diseased) using CheXpert (Irvin et al., 2019) and MIMIC-CXR (Johnson et al., 2019) (**CXR**), (iii) equipment, prevalence, and geographic location changes for diabetic retinopathy assessment models, combining multiple public fundus imaging datasets (Karthik & Sohier, 2019; Decencière et al., 2014; Dugas et al., 2015) (**RETINA**), and (iv) staining protocols changes in histopathology, using **WILDS-CameLyon** (Koh et al., 2021). Finally, we test against hard modality shifts in natural image classification using **DomainNet** (Peng et al., 2019) with 'Real' images as the ID domain. Details about datasets and ID/shifted domain definitions can be found in Appendix A.1, including visual examples of samples from both ID and shifted test sets for each dataset.

## 3.2 Models

Experiments are performed in two settings. First, for models trained with randomly initialised weights, where for each dataset, we trained classifiers with six different architectures: ResNet18/50 (He et al., 2016), EfficientNet (Tan & Le, 2019), MobileNet (Howard et al., 2017), ViT-Base (Dosovitskiy et al., 2020), ConvNext-tiny (Liu et al., 2022), keeping the checkpoint with the highest balanced accuracy on the validation set. We additionally investigate the calibration of models finetuned from pretrained foundation models, using the vision encoders from Dino-V2-Base (Oquab et al., 2023), CLIP (Radford et al., 2021), SiG-LIP (Zhai et al., 2023), MAE-Base (He et al., 2021) and, for medical datasets only, BioMedClip (Zhang et al., 2025). Additional details regarding our training setup and hyperparameters can be found in Appendix A.4. All detailed training configurations are also available in our codebase to ensure reproducibility.

## 3.3 Benchmarked methods

In this study, we aim to analyse the robustness of existing post-hoc calibration methods and their interactions with in-training calibration strategies, on a large variety of shifts and tasks. In terms of post-hoc methods, we compare **TS** (Guo et al., 2017), **IROVaTS** (Zhang et al., 2020), **IRM** (Zadrozny & Elkan, 2002), and **EBS** (Kim & Kwon, 2024). EBS has been shown to achieve state-of-the-art performance on calibration under synthetic corruptions for CIFAR and ImageNet (Kim & Kwon, 2024). However, one crucial difference separates EBS from other calibrators: it requires exposure to a small 'semantic OOD' set for calibration. This set is *unrelated* to the classification task and independent of the tested modality/imaging domain. Following Kim & Kwon (2024), we use samples from the Textures (Cimpoi et al., 2014) dataset for this purpose. However, we consider that comparing 'standard' post-hoc calibrators, such as TS, IRM, etc., with EBS is 'unfair' since EBS gets some outlier exposure in the calibration step, and, by default, the others don't. Hence, to better understand whether any potential calibration gains from EBS should be attributed to the energy-based calibration approach or the use of semantic OOD data, we also compare with augmented versions of these standard post-hoc calibrators where the same semantic OOD set is used along with the ID calibration set in the calibration stage. This yields additional calibrators **TS + OOD**, **IRM + OOD**, **IROVaTS + OOD** (see Appendix A.5.2 for implementation details). Conversely, we also report results of energy-based calibration in the absence of semantic OOD exposure, referred to as **EBS-**. Finally, we investigate the impact of **DAC** (Tomani et al., 2021), designed to improve calibration under shift, in combination with various calibrators.

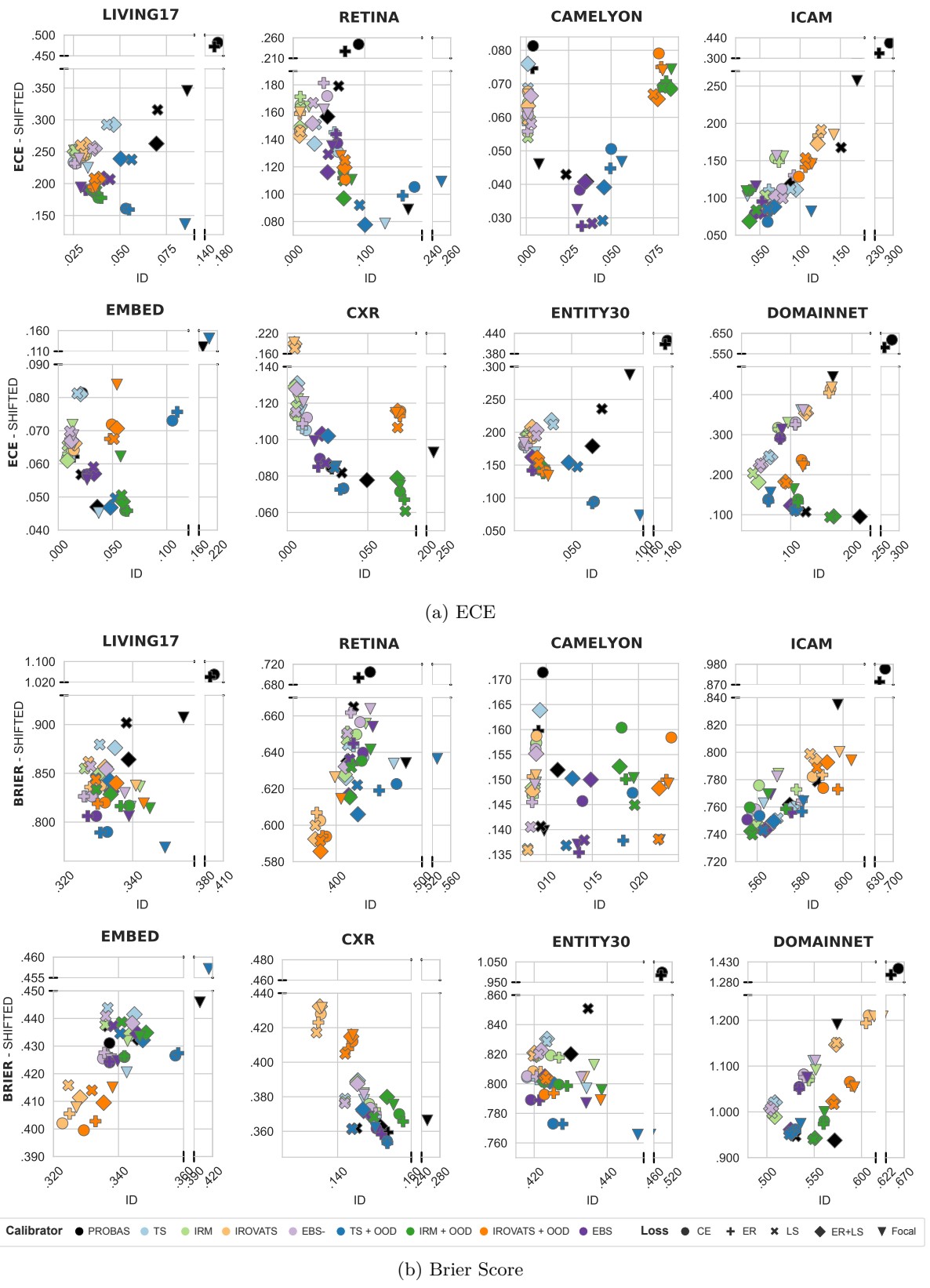

Figure 3: **Calibration under shifts in function of in-distribution calibration for classifiers with random weight initialisation:** For each dataset, we report average ECE (top) and Brier score (bottom), averaged over different models (6 architectures).

In terms of training strategies, we compare standard expected risk minimisation training with cross-entropy loss (**CE**), label smoothing (**LS**), entropy regularisation (**ER**), and **focal** loss. Moreover, given the symmetric effects of label smoothing and entropy regularisation, we also compare with a training strategy using both label smoothing and entropy regularisation simultaneously (**ER+LS**). We then analyse interactions between training strategies and post-hoc calibration methods. In terms of regularisation strength, we chose to fix the hyperparameters for the strength of label smoothing and entropy regularisation across all datasets and experiments ($\lambda = 0.05$ for LS, $\alpha = 0.1$ for ER, $\lambda = 0.05, \alpha = 0.1$ for ER+LS). Fixing these hyperparameters was a deliberate choice (chosen based on the best average ID calibration results across all datasets) to assess the generalisability of a given training strategy to new scenarios. Indeed, as we don't have access to the shifted test set in advance, we would not be able to tune those precisely for improving shifted calibration.

## 4 Results

We evaluate calibration using Expected Calibration Error (ECE) as well as the Brier Score. Whereas ECE focuses on top-label calibration, the Brier score measures full calibration and has the advantage of also taking model performance into account, offering two complementary measures of model calibration (see Section 2.1). In Fig. 3 (resp. Fig. 6), we report both ID and shifted calibration results for every combination of training loss and post-hoc calibration methods, for every dataset, for models trained from scratch (resp. finetuned from foundation models). In the following, we summarise the main take-aways from these experiments.

### 4.1 Calibration under shifts for models trained from scratch

### 4.1.1 Post-hoc calibration: exposure to semantic OOD significantly improves calibration under shift.

In Fig. 3, calibrators without semantic OOD exposure appear in lighter colours whereas their counterparts with exposure appear in darker colours. First, we notice that *after* post-hoc calibration, there is no consistent difference across the various tested training losses – no calibration-specific training loss consistently beats the standard cross-entropy loss. Most strikingly, across all datasets, we observe that calibrators with OOD exposure offer the best results in terms of calibration under shifts (y-axis), for both ECE and Brier scores, regardless of the training loss used. We, for example, find that surprisingly, simple temperature scaling with semantic exposure (TS + OOD) performs the most consistently across datasets and is at least as good as EBS, which was specifically designed for tackling calibration under shifts. Note that this improvement in calibration occurs even though the OOD data used is entirely unrelated to the task, and fixed across all evaluated datasets (Textures dataset, see Section 3.3). This shows that semantic OOD data does not need to be close to the ID images to benefit calibration. Some datasets, like Living17, are closer to images in the Textures dataset (RGB photos) while medical image datasets are further. In Fig. A.3, we confirm this by analysing differences between samples from the ID test set, from the shifted test sets and from the semantic OOD Textures in the embedding space across all datasets, showing that for some datasets the semantic OOD samples are much closer to the ID/shifted test sets than for others. However, results in Fig. 3 show that, regardless of the distance of the semantic OOD set to the ID data, adding exposure to these OOD images benefits calibration under shifts.

After statistical analysis, we confirm that calibrators with semantic OOD exposure have a significantly different effect over those without OOD exposure, however there are no significant differences across calibrators once controlling for semantic OOD exposure (see Appendix A.7 for details on statistical analysis). Regarding model performance, it is important to highlight that most post-hoc calibrators tested here are accuracy-preserving (TS, IRM, EBS), i.e. they do not affect the class prediction. Only IROVaTS affects model predictions and we note that it tends to worsen balanced accuracy both ID and under shifts, across datasets (see Fig. A.4).

Regarding the amount of semantic data to use for calibration, authors of EBS used pre-defined numbers of OOD images for their experiments on CIFAR/ImageNet, but no rationale was given on how to choose this number on new datasets. We found that the number of OOD samples has to be dependent on the size of the ID calibration set. In Fig. 4, we provide a detailed investigation of the sensitivity of various calibrators to the relative size of the OOD set. Results show that the ideal OOD size depends on the dataset, nevertheless,

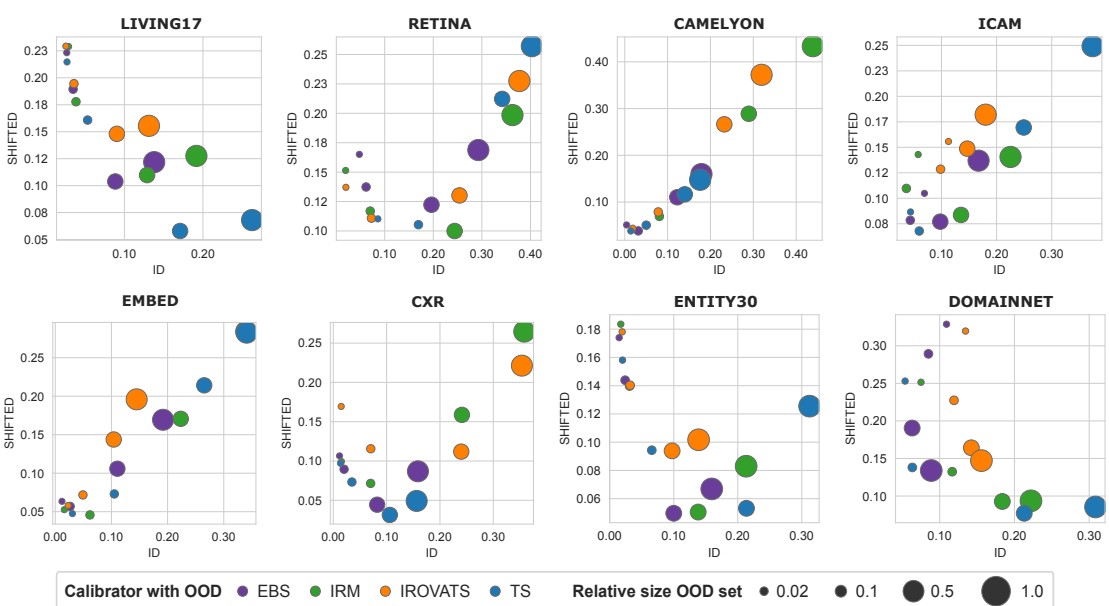

Figure 4: **Effect of the amount of semantic OOD data used for calibration on ECE:** (averaged over six model architectures). The amount of OOD data is here relative to the number of ID validation samples used to calibrate the model (varying across datasets).

we find that when the semantic OOD set is about 10% the size of ID validation set (used for calibration), the calibration performance lies on the Pareto front of ID-shifted calibration for most post-hoc calibrators and datasets. We chose this parametrisation across all experiments.

Finally, we additionally evaluate the effect of complementing post-hoc calibrators with Dynamically Aware Calibration (DAC). DAC uses distances in the embedding spaces to further optimise the sample-wise temperature and is compatible with many calibrators. Results in Fig. 5 suggest that DAC has mixed effects on model calibration, with substantial gains for some datasets (EMBED) and worsening calibration under shifts for others (IROVaTS/EBS CameLyon). Overall, it simply has little effect on many datasets (Living17, CXR, Entity30).

### 4.1.2 ER+LS training yields the best calibrated raw outputs on shifted data and is hard to beat with post-calibrators.

Without post-hoc calibration (black markers), models trained with both entropy-regularisation and label smoothing (ER+LS, diamond) are best calibrated, often by a large margin, both in terms of ECE and Brier score. For example, on Living17 and Entity30 the ECE is 50% lower for models trained with ER+LS than with CE. Overall, without post-hoc calibration, ER+LS has the lowest ECE across all tested training paradigms on 7 out of the 8 tested datasets, and the lowest Brier in 6 out of 8. Importantly, nor LS nor ER alone is able to reach the same consistent calibration robustness, combining both paradigms is key. Importantly, we find that the different loss functions perform similarly in terms of balanced accuracy across datasets (see Fig. A.4).

### 4.1.3 Improving calibration under shift often comes at the cost of ID calibration.

In Fig. 3, we observe that on several datasets (Living17, RETINA, CameLyon, CXR, Entity30) a trade-off pattern between ID and shifted calibration appears, in particular for ECE. This is particularly striking for calibrators with OOD exposure, which yield much lower calibration errors under shifts compared to other calibrators, but sometimes display substantially higher calibration error on ID test sets.

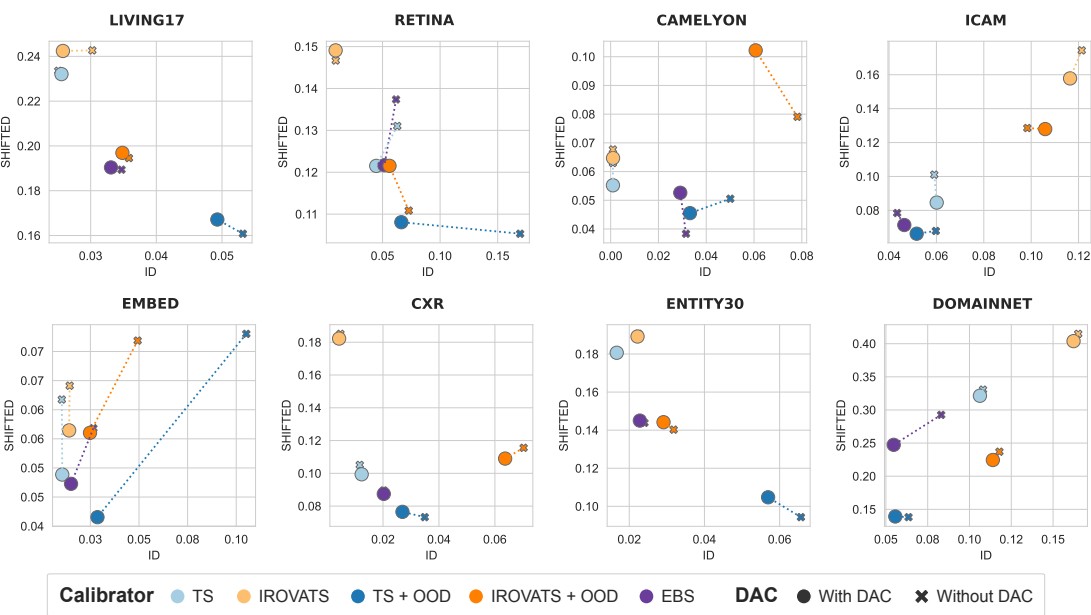

Figure 5: **Effect of DAC with various post-hoc calibrators on ECE: mixed results across datasets**. Large circles show calibration with DAC; crosses denote calibration without DAC. Full comparison with Brier score can be found in Appendix A.8.

### 4.2 Do these findings hold for classifiers finetuned from foundation models?

#### 4.2.1 Findings from randomly initialised models generalise to models finetuned from foundation models

Fig. 6 compares calibration results both under shifts and on ID test sets, for classifiers finetuned from foundation models (averaged over four to five models, see Section 3.2). These results confirm our previous conclusions: exposure to semantic OOD improves calibration under shift, with EBS and TS+OOD yielding most consistent results in terms of ID-shifted trade-off across datasets (average ID-shifted calibration is best for TS+OOD on 6 out of 8 datasets). In terms of base predictions (without post-hoc calibration), models finetuned with ER+LS are best calibrated under shifts compared to base predictions from models trained with other losses (black markers, y-axis) on 6 out of 8 datasets, however this again often comes at the cost of ID calibration (3 out 6 datasets).

#### 4.2.2 Models finetuned from foundation models are better calibrated under dataset shifts, irrespective of the calibrator used

In Fig. 6, the red star depicts average calibration for models trained from scratch (with CE) and using robust EBS post-hoc calibration, a strong baseline from Fig. 3. This comparison shows that models finetuned from foundation models offer substantially better calibration under dataset shifts, in terms of Brier score in particular (also reflecting an improvement in classification performance, see Fig. A.5), but also in terms of ECE. This conclusion may perhaps not be very surprising per se, however, it is important to note that even with foundation models applying robust post-hoc calibration like EBS or TS with semantic OOD significantly improves calibration compared to base probabilities. Moreover, a further ablation study in Fig. A.8 on the impact of model size on shifted-ID calibration demonstrates the calibration improvements are indeed attributable to the pretraining of the foundation models and not to their increased model size.

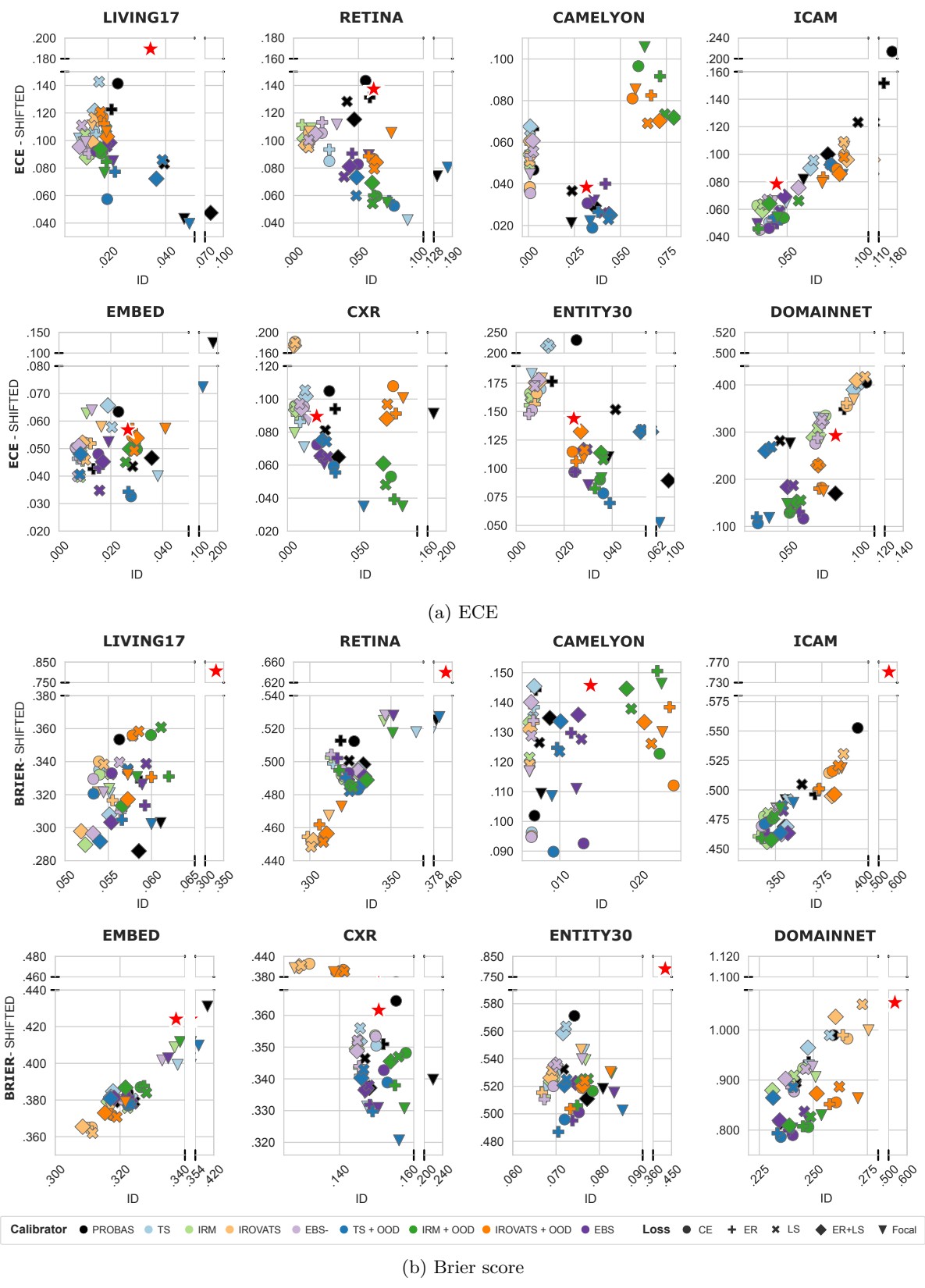

Figure 6: **Calibration results for classifiers finetuned from foundation models:** top row ECE, bottom row Brier Score (average calibration finetuning four different foundation models, five for medical datasets). As a baseline, for each dataset, the red star depicts the average calibration of models trained from scratch with CE loss and EBS calibration.

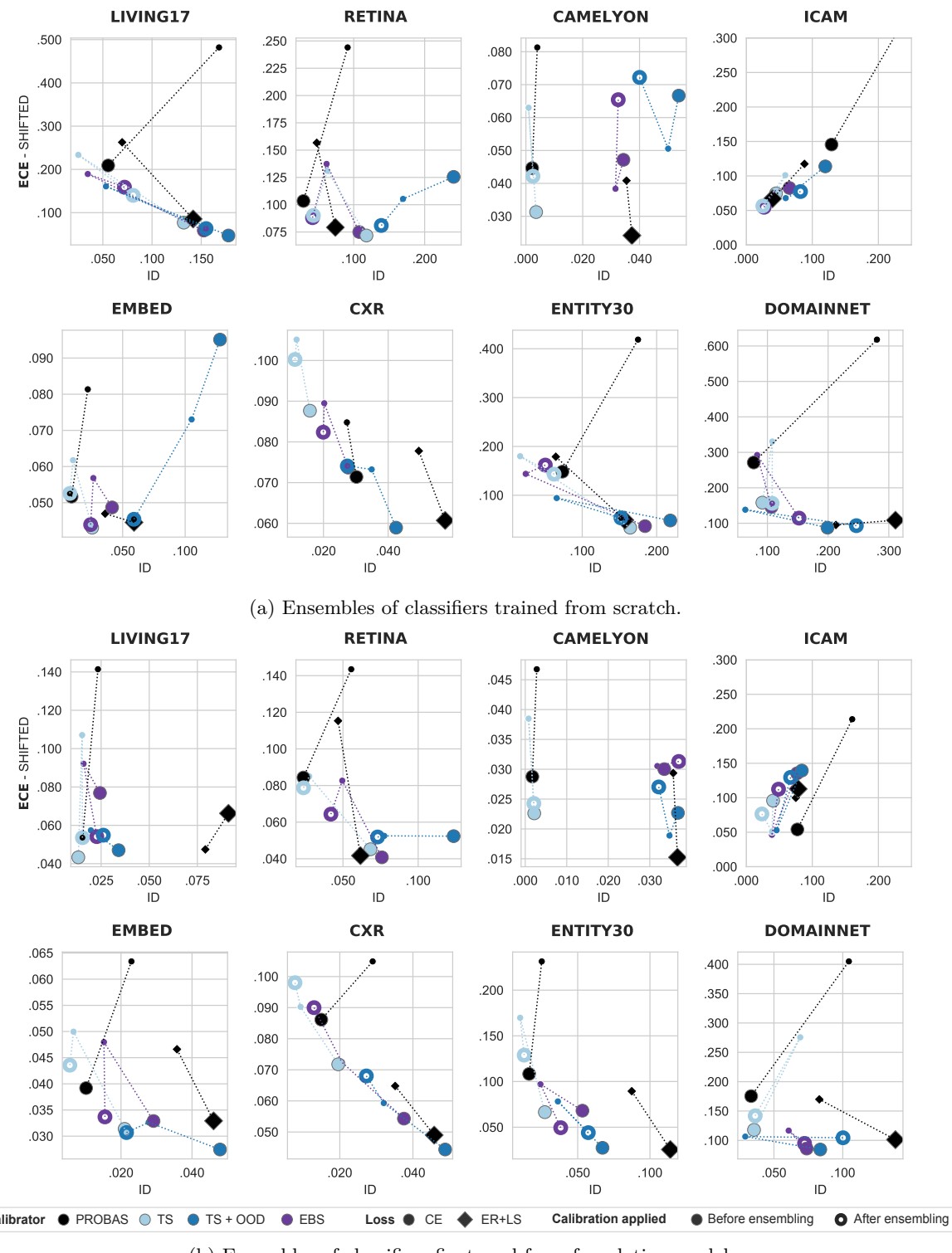

(a) Ensembles of classifiers trained from scratch.

(b) Ensembles of classifiers finetuned from foundation models.

Figure 7: **Calibration of model ensembles in terms of ECE, in function of post-hoc calibration method**. Large dots denote calibration results for model ensembles, small dots the average calibration of individual ensemble members for reference. We compare the effect of (i) applying post-hoc calibration to ensemble members (before ensembling), (ii) applying post-hoc calibration after ensembling predictions.

### 4.3 Ensembles and shifted calibration

In light of conflicting results on the effect of model ensembling on calibration in the literature (see Section 2.5), we here investigate whether the previous findings experimentally generalise to model ensembles. For each dataset, we create various model ensembles by sampling random combinations of three models from the pool of previously trained classifiers. We investigate two types of ensembles: ensembles of classifiers trained from random initialisation and ensembles of classifiers finetuned from foundation models. In Fig. 7, we compare calibration results for different post-hoc calibrators when calibration is applied: (i) to individual ensemble members before ensembling or (ii) after ensembling (as argued in (Rahaman & Thiery, 2021)). Motivated by take-away 2, we additionally compare with ensembles of models trained with ER+LS.

#### 4.3.1 Ensembling remains a very effective method for calibration

Fig. 7 shows that ensembles achieve better calibration under shifts compared to individual models (small versus large dots), irrespective of the calibration strategy. This is visible both in ECE and even more strikingly in the Brier score (also reflecting, as expected (Lakshminarayanan et al., 2017), an improvement in model performance after ensembling). Then, comparing results in Figs. 7a and 7b, we notice that calibration results lie in drastically different scales for ensembles of classifiers trained from scratch and ensembles of classifiers finetuned from foundation models. The latter being much better calibrated overall. Finally, in Figs. A.12 to A.15, we investigate whether these calibration gains depend on the size of the ensemble $n$, by comparing $n \in \{2, 3, 5\}$. We observe that calibration gains are already visible with ensembles with two members only. Gains on shifted calibration increase slightly as the number of members increases, however there is a plateau-effect with only minimal improvements when increasing from 3 to 5 ensemble members.

#### 4.3.2 Applying post-hoc calibration *before* ensembling yields best calibration under shifts.

Comparing post-hoc calibration pre- and post-ensembling, results in Fig. 7 show that there is a clear trade-off between ID and shifted calibration in terms of ECE: doing TS (or EBS) after ensembling leads to better ID calibration, whereas TS (or EBS) before ensembling leads to better calibration on shifted datasets, and better overall ID-shifted calibration trade-off. For example, TS before ensembling (filled circles) systematically reaches a lower calibration error on shifted test sets compared to TS after ensembling (hollow circles). However, we observe the inverse pattern for ID test sets on Living17, Density, Retina, CXR, and Entity30. This finding also holds for models trained with ER+LS (see Fig. A.11). Finally, our ablation study on ensemble size in Figs. A.12 to A.15 shows that these conclusions hold for all ensemble sizes tested ($n = 2, 3, 5$).

Comparing TS and TS + OOD in Fig. 7 we find that semantic OOD exposure in post-hoc calibration often degrades calibration for shifted tests (y-axis, RETINA, EMBED, CameLyon) and even more so for ID test sets (x-axis). Interestingly, this is contrary to previous findings and independent of whether calibration is applied before or after ensembling. Further ablation studies in Fig. A.10 confirm this finding on additional post-hoc calibrators. These results suggest that, if the goal is to ensemble models, individual ensemble members should be calibrated with standard calibration methods, not with OOD exposure.

#### 4.3.3 Without post-hoc calibration, ensembles whose members were trained with ER+LS are substantially better calibrated under shifts.

Finally, comparing ensemble calibration without post-hoc calibration (black markers in Fig. 7), we can, again, see that ensembles whose members were trained with ER+LS offer substantially better calibration results under shifts compared to ensembles whose members were trained with CE, across all datasets, for both ECE and Brier score. However, this often comes at the cost of worse ECE on ID test sets (Living17, Entity30, EMBED, CXR, DomainNet).

# 5    Conclusion

We conduct a large-scale experimental evaluation of calibration methods in the presence of realistic dataset shifts. Associated findings provide practical guidelines for constructing models with robust calibration under such shifts. First, for single models (i.e. without ensembling), we find that post-hoc calibrators exposed to task-agnostic OOD data performed best for calibration under shifts, particularly EBS and TS+OOD. This finding holds irrespective of the training paradigm, across datasets, for classifiers trained from scratch and finetuned from foundation models. Yet, this can come at the cost of worsening ID calibration, yielding an ID-shifted calibration trade-off. Moreover, and perhaps surprisingly, this finding does not hold for model ensembles, where we find that OOD exposure should *not* be used for calibration. Moreover, we find that models trained with entropy regularisation and label smoothing provided the best calibrated base probabilities for shifted test sets (without any post-hoc calibrator) across datasets.

Based on these findings, we derive practical guidelines for robust calibration under shifts, visually summarised in Fig. 2. First, in terms of training strategy: the best results are obtained when classifiers are initialised with foundation model weights (as opposed to random initialisation). Then, regarding the use of in-training calibration techniques, we find that they are not needed if post-hoc calibration is used. However, if the aim is to bypass post-hoc calibration, we find in-training calibration with ER+LS to be most effective. Regarding the choice of post-hoc calibrators, we find that applying TS+OOD (or EBS) is a robust strategy for improving calibration under shifts of single classifiers (without ensembling). Finally, regarding model ensembles, we find that simply calibrating ensemble members with TS *prior* to ensembling yields robust calibration results. We find that combining this ensembling strategy with finetuning from foundation models yields the most robust calibration results overall, both ID and under shifts.

In this study, we focused exclusively on robustness of calibration under dataset shifts for image classification. It is worth noting that building reliable and well-calibrated uncertainty estimates, robust across dataset shifts, is equally relevant for other tasks. Gustafsson et al. (2023) for example studied the reliability of existing uncertainty estimates in the context of regression tasks, whereas Jorge et al. (2023) similarly studied robustness and calibration of uncertainty estimates in segmentation tasks. Moreover, the guidelines proposed here are based on our extensive evaluation of currently commonly used post-hoc and in-training calibration methods, and these may require updating as new calibration methods appear in the future. To this end, our publicly available benchmarking codebase provides a comprehensive evaluation framework and facilitates future benchmarking efforts for calibration under shifts in image classification.

### Acknowledgments

M.R. is funded by an Imperial College London President's PhD Scholarship and a Google PhD Fellowship. R.M. is funded by the European Union's Horizon Europe research and innovation programme under grant agreement 101080302. B.G. and F.d.S.R. acknowledge the support of the UKRI AI programme, and the EPSRC, for CHAI - EPSRC Causality in Healthcare AI Hub (grant no. EP/Y028856/1). B.G. received support from the Royal Academy of Engineering as part of his Kheiron/RAEng Research Chair.

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

## A.1   Dataset definitions

In this work we study realistic shifts in the medical imaging and natural images domain. For this we leverage multiple public datasets across various modalities and tasks. We here detail how we defined ID and OOD domains for non-standard benchmark datasets. We summarise splits and ID/shifted domains definitions in Table A.1. In Figs. A.1 and A.2 we illustrate the types of shift by showing examples of images from both the ID and the shifted sets for each dataset.

| Dataset | Type of shift(s) | N classes Balanced (Y/N) | ID domain $(N_{train}, N_{val}, N_{test})$ | Shifted domains (N images) |
|---------|------------------|--------------------------|---------------------------------------------|-----------------------------|
| CXR | Scanner Prevalence Population | 2 classes (N) | CheXpert random train/val/test split (129732, 23086, 38192) | MIMIC-CXR random subset (25000) |
| RETINA | Scanner Prevalence Population | 5 classes (N) | EyePACS (35126, 10715, 42861) | Messidor (1744) Aptos (3662) |
| EMBED | Scanner | 4 classes (N) | Selenia Dimension 2D (169758, 43055, 52975) | Senographe Essential (12218) Senograph 2000D (13268) Lorad Selenia (10515) Senographe Pristina (498) Clearview CSm (8128) Selenia Dimensions C-views (108369) |
| CameLyon | Staining | 2 classes (Y) | Official ID splits Hospitals 1,2,3 (272192, 30244, 33560) | Official OOD splits Hospital 4 (34904) Hospital 5 (85054) |
| iCam | Geographic Prevalence | 182 classes (N) | Official ID splits (129809, 7314, 8154) | 'OOD val' split (14961) 'OOD test' split (42791) |
| Living17 | Population | 17 classes (Y) | Source domain from official 'rand' split (37570, 6630, 1700) | Target test set from official 'rand' split (1700) |
| Entity30 | Population | 30 classes (Y) | Source domain from official 'rand' split (131123, 23140, 6000) | Target test set from official 'rand' split (6000) |
| DomainNet | Image type Prevalence | 345 classes (N) | Official ID splits from Wilds with ID domain 'Real' (217630, 24182, 104082) | Clipart test (14604) Infograph test (15582) Painting test (21850) Quickdraw test (51750) Sketch test (20916) |

Table A.1: Datasets and shifts used in this study, details on ID and shifted splits definition.

**EMBED**   we analyse robustness against scanner changes for breast density assessment models in mammography. For this purpose, we use the public available EMBED dataset (Jeong et al., 2023). This dataset contains data from 6 different scanners (Selenia Dimensions, Senographe Essential, Senograph 2000D, Lorad Selenia, Clearview CSm, Senographe Pristina). In this work, we used Selenia Dimensions (2D images) as the ID scanner for training and validation, and all the other scanners, as well as the C-view images from Selenia Dimensions, are used as shifted test sets, resulting in six shifted test sets.

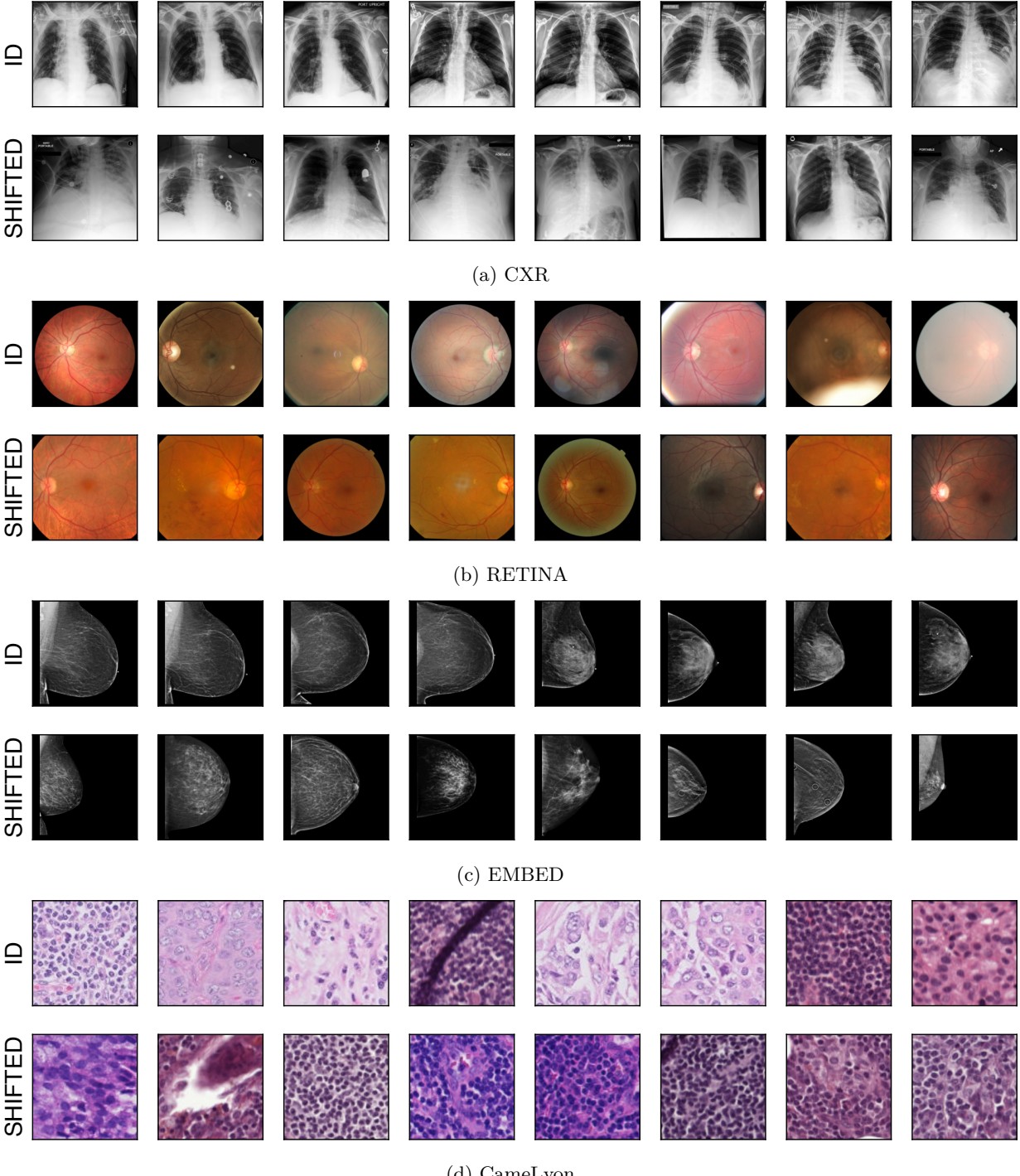

(a) CXR

(b) RETINA

(c) EMBED

(d) CameLyon

Figure A.1: Illustration of each dataset and associated shifts (1/2).

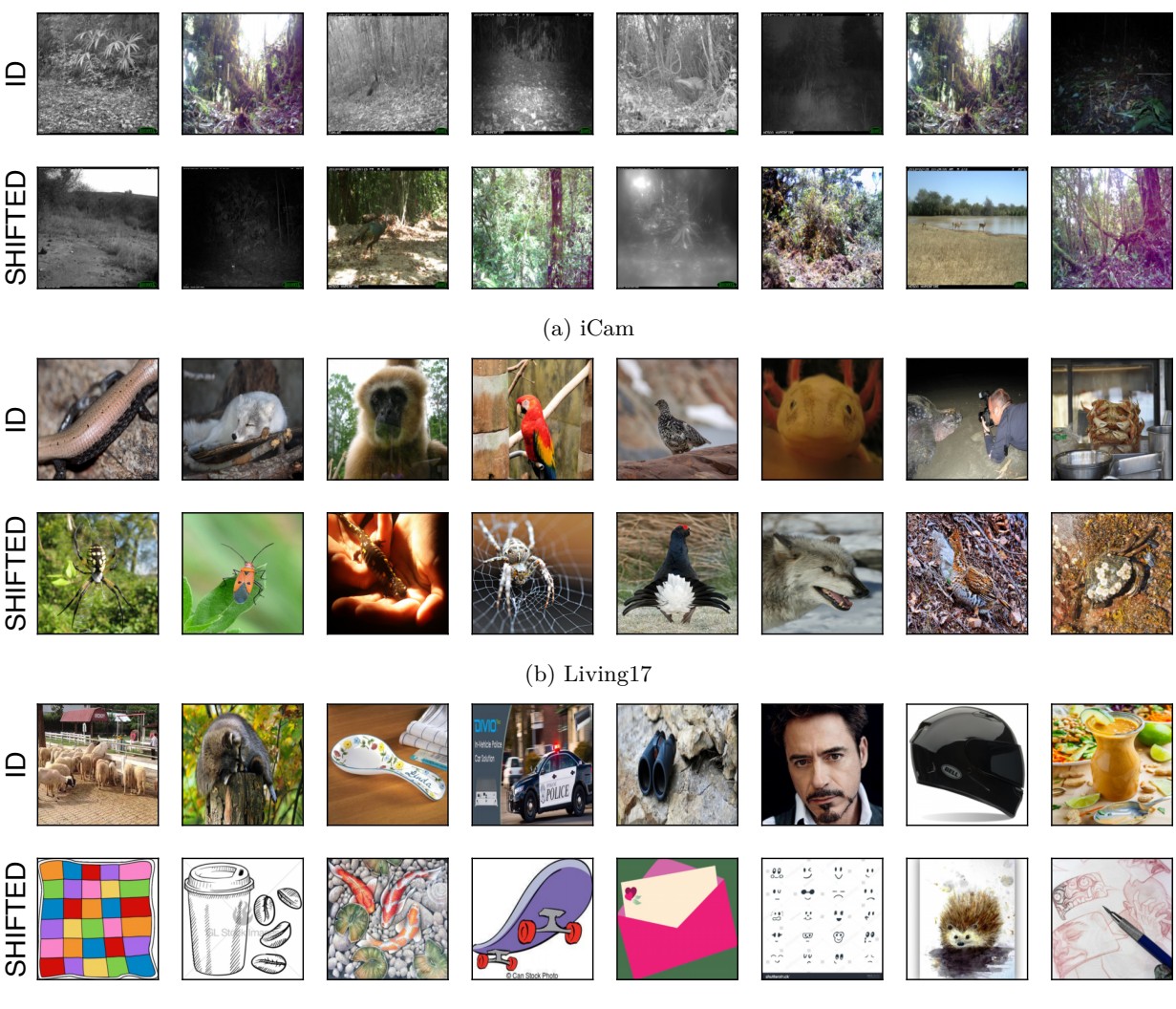

(a) iCam

(b) Living17

(c) DomainNet

Figure A.2: Illustration of each dataset and associated shifts (2/2).

**RETINA** we analyse robustness against equipment and geographic location changes in diabetic retinopathy grading models by combining three publicly available datasets: Kaggle Diabetic Retinopathy Challenge (EyePACS) dataset (Dugas et al., 2015), Kaggle Aptos dataset (Karthik & Sohier, 2019) and the Messidor-v2 (Decencière et al., 2014) dataset. These datasets cover various locations (US, India and France) and different level of equipment from high-quality scanners to mobile phone cameras. We use the largest dataset (EyePACS) as ID domain, and images from the two other datasets as OOD test sets.

**CXR** we analyse robustness against scanner, location and label distribution changes in chest-xray classification models (No Finding versus Disease task) using the CheXpert (Irvin et al., 2019) and MIMIC-CXR (Johnson et al., 2019) datasets. We use the CheXpert dataset as our ID domain, splitting randomly between train/val/test at the patient level, and we use a random subset of 25,000 images from the MIMIC-CXR database as our OOD test set.

**DomainNet** we use the implementation and splits definition provided by the Wilds package (Koh et al., 2021) for our DomainNet (Peng et al., 2019) experiments. Specifically, we use images from the 'REAL' domain as our ID images (train/test/test), and images from all other 5 domains (clipart, infograph, painting, quickdraw, sketch).

## A.2 Details on calibration metrics computation

Expected Calibration Error, or ECE (Guo et al., 2017) approximates the expected difference between predicted confidence and prediction accuracy. We follow the notations from the main paper: with a classification task with $C$ classes, $x \in \mathcal{X}$ the model input, and $y \in \{1, \ldots, C\}$ the associated label. Let $\hat{\mathbf{p}} \in [0,1]^C$ denote the model predicted probabilities for each class, $\hat{y} = \arg\max_i \hat{p}_i$ the predicted class, and $\hat{c} = \max \hat{p}_i$ the associated confidence score. To compute ECE, the output space $[0,1]$ is partitioned into $M$ bins, for each bin $B_m$, we compute the average confidence score in the bin $\text{Conf}(B_m) = \frac{1}{|B_m|} \sum_{\hat{c}^i \in B_m} \hat{c}^i$ and the actual accuracy in this bin $\text{Acc}(B_m) = \frac{1}{|B_m|} \sum_{\hat{c}^i \in B_m} \mathbb{1}[y^i = \hat{y}^i]$. The ECE then reflects the average error over all bins, weighted by the number of samples in each bin:

$$ECE = \sum_{m=1}^{M} \frac{|B_m|}{N} |\text{Acc}(B_m) - \text{Conf}(B_m)| \tag{1}$$

with $N$ the total number of samples.

The Brier Score (Brier, 1950), on the other hand computes the average class-wise mean squared error between probabilities and one-hot labels. Formally, the score is computed as: $\frac{1}{N} \sum_{i=1}^{N} \sum_{k=1}^{C} \left( \hat{p}_k^i - y_k^i \right)^2$.

## A.3 Comparison with existing studies

For the convenience of the reader, below we provide a comparative table summarising differences between previous studies on calibration under shifts in image classification and our study.

| Paper | Datasets | Type of shifts | Post-hoc methods | In-training calibration methods | Bayesian approaches | Effect of ensembling | Effect of foundation models |
|---|---|---|---|---|---|---|---|
| Ovadia et al. (2019) | Rotated MNIST, Translated MNIST, CIFAR10-C, ImageNet-C | Synthetic | TS | - | ✓ | ✓ | - |
| Tomani et al. (2021) | CIFAR10-C, ImageNet-C, ObjectNet | Synthetic | TS, IR, IRM, TS-IR with(out) perturbation on val set | - | - | - | - |
| Tomani et al. (2023) | CIFAR10-C, CIFAR100-C, ImageNet-C | Synthetic | TS, ETS, IRM, SPL with(out) DAC | - | - | - | - |
| Kim & Kwon (2024) | CIFAR10-C, CIFAR100-C, ImageNet-C | Synthetic | ETS, SPLINE, EBS, IRM, IROVa, IROVaTS, DAC | - | - | - | - |
| Yu et al. (2022) | ImageNet-C, WILDS-RxRx1, GLDv2 | Synthetic + Real-world | TS, MD+TS | - | - | - | - |
| Kumar et al. (2022) | Cropland, Landcover, CelebA, Living17, Entity30, DomainNet, ImageNet, iWild-Cam, Waterbirds | Real-world | TS | - | ✓ | - | - |
| Zhang et al. (2022) | CIFAR10-C, CIFAR100-C, TinyImageNet-C | Synthetic | TS | LS, Focal | - | - | - |
| Seligmann et al. (2023) | WiLDS datasets (iCAM, FMoW, RxRx1), CIFAR10-C, | Synthetic + Real-world | - | - | ✓ | ✓ for BNNs | - |
| **Ours** | WiLDS datasets (iCam, CameLyon), Living17, Entity30, EMBED, CheXpert, MIMIC-CXR, RETINA, DomainNet | Real-world | TS (+OOD), IROVaTS (+OOD), IRM (+OOD), EBS | LS, ER, ER+LS, focal | - | ✓ | ✓ |

Table A.2: Comparison of our and other existing studies on calibration under dataset shifts in image classification, see Section 2 for more details.

| Dataset | LR schedule for CNNs | LR schedule for ViT | LR schedule for ConvNext | Max epochs |
|---------|---------------------|---------------------|--------------------------|------------|
| Living17 | Cosine $10^{-3}$ to $10^{-5}$ | Fixed $10^{-5}$ | Fixed $10^{-4}$ | 50 |
| Entity30 | Cosine $10^{-3}$ to $10^{-5}$ | Fixed $10^{-5}$ | Fixed $10^{-4}$ | 100 |
| CXR | Cosine $10^{-4}$ to $10^{-5}$ | Fixed $10^{-5}$ | Fixed $10^{-4}$ | 50 |
| Density | Cosine $10^{-4}$ to $10^{-5}$ | Fixed $10^{-5}$ | Fixed $10^{-4}$ | 50 |
| iCam | Cosine $10^{-4}$ to $10^{-5}$ | Fixed $10^{-5}$ | Fixed $10^{-4}$ | 100 |
| RETINA | Cosine $10^{-3}$ to $10^{-5}$ | Fixed $10^{-5}$ | Fixed $10^{-4}$ | 150 |
| DomainNet | Cosine $10^{-3}$ to $10^{-5}$ | Fixed $10^{-5}$ | Fixed $10^{-4}$ | 150 |
| CameLyon | Cosine $10^{-3}$ to $10^{-5}$ | Fixed $10^{-5}$ | Fixed $10^{-4}$ | 150 |

Table A.3: Learning rate schedules used to train models initialised with random weights, per dataset. Learning rate schedules were chosen based on best validation performance.

## A.4 Hyperparameters and training setup details

All of our classification models were trained using a standard cross-entropy loss, using the Adam optimiser, with a batch size of 32. For models trained from scratch, we used a cosine learning rate schedule or a fixed learning rate schedule, depending on the architecture and the dataset, chosen based on validation performance, we detail all configurations in Table A.3. For finetuning foundation models, we used a fixed learning rate of $10^{-5}$, chosen based on validation performance. We performed full-finetuning of foundation models, i.e. all weights were finetuned. For all experiments, we chose the checkpoint with the highest validation balanced accuracy for inference.

## A.5 Post-hoc calibrators with semantic OOD exposure: implementation details

### A.5.1 EBS: Energy-based calibration, implementation details

In this section, we detail the algorithm proposed by Kim & Kwon (2024) for their energy-based calibration method.

---

**Algorithm 1:** Energy-based calibration (Kim & Kwon, 2024)

---

**Input:** In-distribution dataset: $\mathcal{D}_{id} = \{(\mathbf{x}_i, y_i)\}_{i=1}^N$
**Input:** Classifier $f(\mathbf{x})$
**Input:** Semantic OOD data: $\mathcal{D}_{out} = \{(\tilde{\mathbf{x}}_i, 0^C)\}_{i=1}^{N_{ood}}$
**Input:** Temperature value from temperature scaling: $T_{ts}$
/* Get energy for all samples                                    */
$\mathcal{D} \leftarrow \mathcal{D}_{id} \cup \mathcal{D}_{out}$
$\mathbf{z} \leftarrow f(\mathbf{x})$
$\mathcal{F}(\mathbf{z}) \leftarrow -\texttt{logsumexp}(\mathbf{z})$
/* Estimate Gaussian PDFs on correctly classified samples        */
$P_1 \leftarrow \text{fit}(\mathcal{F}(\mathbf{z}_{\text{correct}}))$
/* Estimate Gaussian PDFs on incorrect + OOD samples             */
$P_2 \leftarrow \text{fit}(\mathcal{F}(\mathbf{z}_{\text{incorrect}}))$
/* Fit calibrator                                                */
$T(\mathbf{z}_i) \leftarrow T_{ts} - P_1(\mathcal{F}(\mathbf{z}_i))\theta_1 + P_2(\mathcal{F}(\mathbf{z}_i))\theta_2$
$\hat{\theta} \leftarrow \arg\min_\theta \sum_{i \in \mathcal{D}} \texttt{MSE}(\texttt{softmax}(\mathbf{z}_i/T(\mathbf{z}_i)), y_i)$
**Output:** $\hat{\theta}, T_{ts}, P_1, P_2$

---

### A.5.2  Enhancing other calibrators with OOD exposure

EBS (Kim & Kwon, 2024) requires exposure to a small 'semantic OOD' set for calibration, unrelated to the task at hand. Kim & Kwon (2024) propose to use samples from the Textures (Cimpoi et al., 2014) dataset as OOD samples. To disentangle the contribution of the energy-based calibration approach and the use of semantic OOD towards improved calibration, in this study we evaluate augmented versions of 'standard' post-hoc calibrators, where the same semantic OOD set is used along the ID calibration set in the calibration stage, yielding additional calibrators **TS + OOD**, **IRM + OOD**, **IROVaTS + OOD**. Concretely, we collect an unlabeled sample of OOD data and combine it with the ID calibration set to fit the calibrator of choice. For the calibration set, labels need to be assigned to the OOD samples, in this case we simply assign the uniform distribution i.e. for all samples $y = [1/C, ..., 1/C]$ where $C$ is the number of classes. We illustrate the full process for the example of temperature scaling with OOD, in Algorithm 2. For any other calibrator (IRM, IROVaTS) the process is identical, only the fit step needs to be adapted.

---

**Algorithm 2:** Temperature scaling with semantic OOD

**Input:** In-distribution dataset: $\mathcal{D}_{id} = \{(\mathbf{x}_i, y_i)\}_{i=1}^{N}$

**Input:** Classifier $f(\mathbf{x})$

**Input:** Semantic OOD data: $\mathcal{D}_{out} = \{\tilde{\mathbf{x}}_i\}_{i=1}^{N_{ood}}$

/* Assign uniform distribution as labels for OOD samples        */

**for** $i = 1 \rightarrow N_{ood}$ **do**

$\quad | \quad \tilde{y}_i \leftarrow \frac{1}{C} \cdot \mathbf{1}_C$

**end**

$\mathcal{D}_{out} \leftarrow \{(\tilde{\mathbf{x}}_i, \tilde{y}_i)\}_{i=1}^{N_{ood}}$

/* Combine both datasets                                          */

$\mathcal{D} \leftarrow \mathcal{D}_{id} \cup \mathcal{D}_{out}$

/* Fit calibrator on the combined dataset, here TS               */

$\hat{T} \leftarrow \arg\min_T \sum_{(\mathbf{x},y)\in\mathcal{D}} \text{CrossEntropy}(\texttt{softmax}(f(\mathbf{x})/T), y)$

**Output:** $\hat{T}$

---

### A.5.3 Ablation study: embedding space analysis between ID, shifted and semantic OOD sets

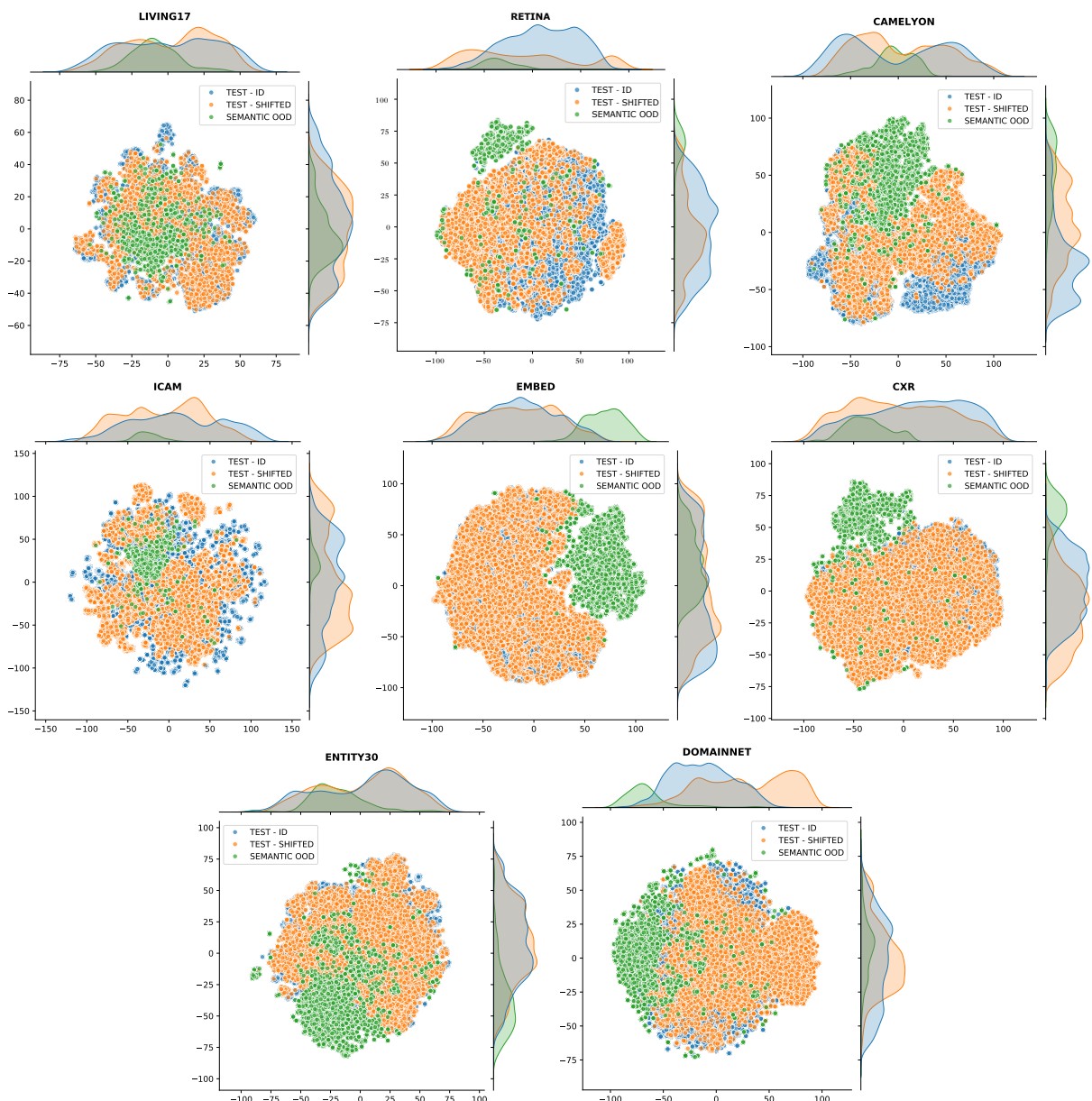

Figure A.3: **Embedding space analysis across datasets with TSNE** (Maaten & Hinton, 2008). For every dataset, we analyse the distribution of the ResNet-18 classifier embeddings from ID test samples, shifted test samples and semantic OOD Textures samples. The TSNE representation is computed based on the entire set (ID + shifted + semantic OOD samples). Note that if either test set contains more than 10,000 samples, we randomly sampled a subset of 10,000 samples.

## A.6 Balanced Accuracy results

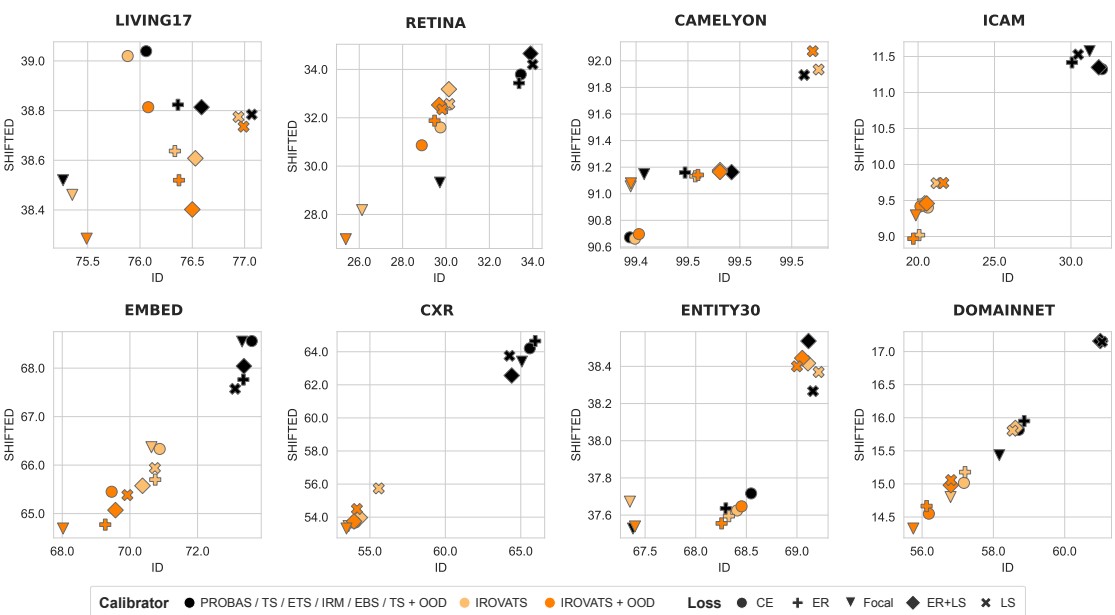

Figure A.4: **Balanced accuracy results (in %) on ID & shifted test sets, models trained from scratch**. Most post-hoc calibrators are accuracy-preserving, however IROVaTS is not and tends to decrease balanced accuracy across datasets. We do not observe any consistent performance differences across training losses.

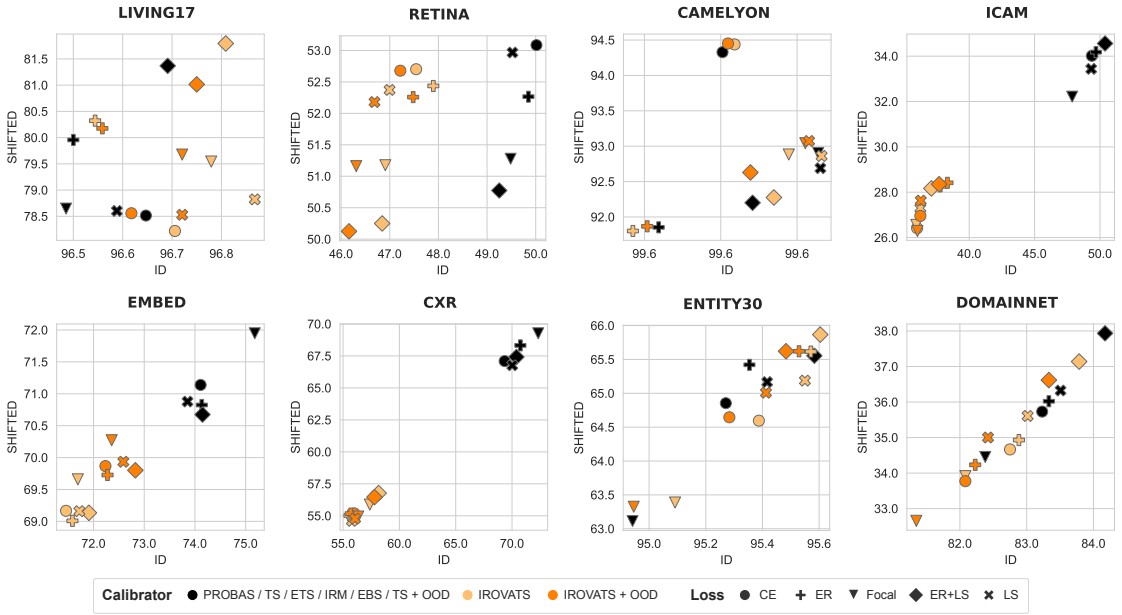

Figure A.5: **Balanced accuracy results (in %) on ID & shifted test sets, models finetuned from foundation models**. Most post-hoc calibrators are accuracy-preserving, however IROVaTS is not and tends to decrease balanced accuracy across datasets. We do not observe any consistent performance differences across training losses.

## A.7 Statistical significance analysis

To determine the significance of the overall training loss + post-hoc calibrator 'treatment' effect across datasets and models, we use the Friedman (Friedman, 1937) and Nemenyi (Nemenyi, 1963) test. Technically, we run this test, separately for ID and OOD test sets. In our input data matrix, each row corresponds to one architecture-dataset combination for which we collect the average calibration error (across available test sets) for every treatment (loss + post-hoc calibrators). We find an overall significant effect with the Friedman test in both scenarios (ID and shifted). Hence, we follow by a Nemenyi post-hoc test to determine which treatment pairs have a significantly different treatment effect on the calibration results, results are presented in Fig. A.6.

We find that on shifted test sets, calibrators with semantic OOD exposure have a significantly different effect over those without OOD exposure, however there are no significant differences across calibrators once controlling for semantic OOD exposure (take-away 1). We also find that base probabilities from ER+LS models are not are not significantly different from any post-hoc calibration with OOD exposure (take-away 2). We find that, for nearly all post-hoc calibrators, after calibration there are no significant differences between different training losses. For ID sets, we find a significant effect of semantic OOD exposure as well, confirming the ID-OOD trade-off detailed in take-away 3.

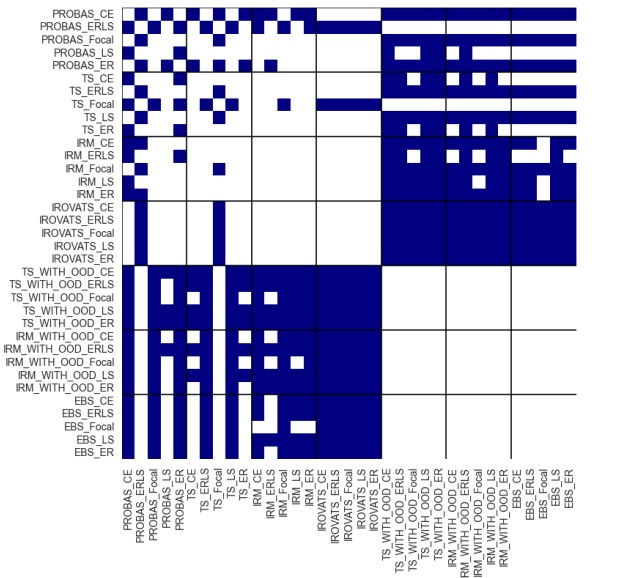
(a) Significant differences on shifted test sets

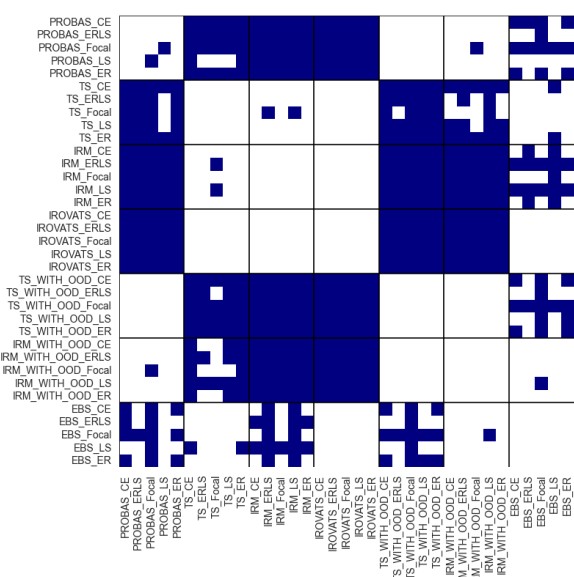
(b) Significant differences on ID test sets

Figure A.6: Full post-hoc comparison results of the Nemenyi test for ECE values on shifted and in-distribution test sets across all datasets and models (trained from scratch). The Nemenyi-post-hoc test identifies which specific treatments are significantly different from each other. Blue indicates a significant differences between the pairs of treatments in the x and y axes.

## A.8    Detailed analysis of DAC effect

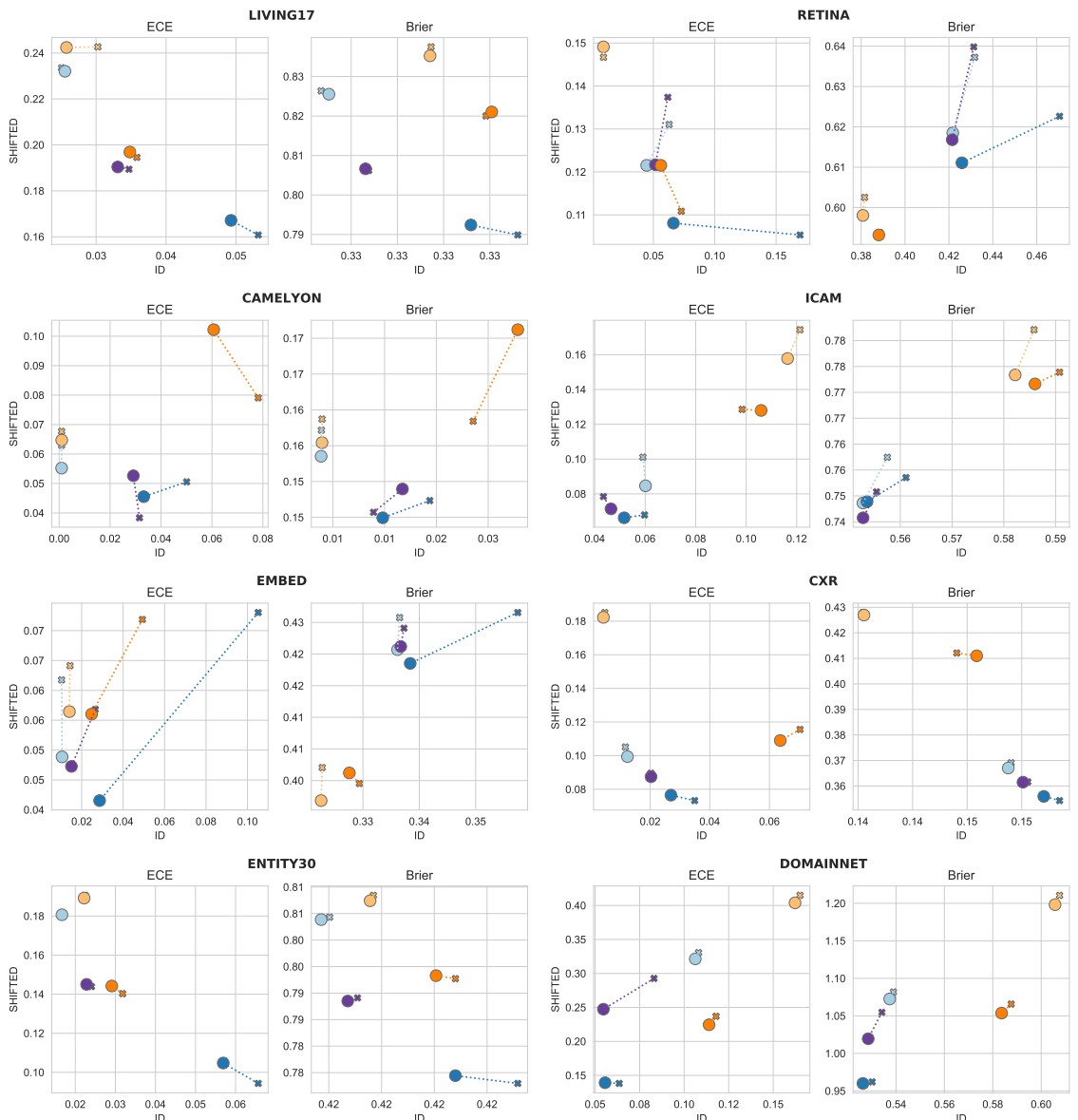

Figure A.7: **Detailed ablation results of DAC effect** in terms of average ECE and Brier score on ID and shifted test sets.

## A.9  Ablation study: effect of model size on shifted calibration

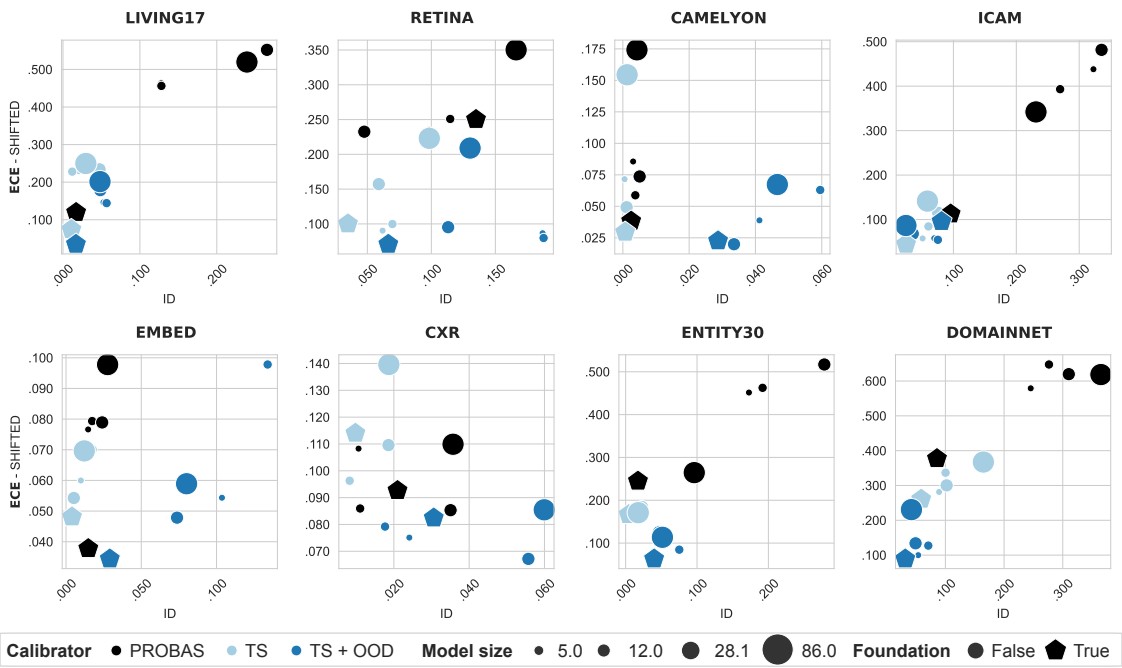

Figure A.8: **Ablation study: calibration results in function of model size**. We plot the ID-shifted ECE results by model size (for models trained from scratch) and compare with VIT-B-DINOv2 (i.e. finetuned from foundation model). We can see that calibration tends to degrade as model size increases. However, findings like benefit of semantic OOD exposure on shifted calibration and ID/shifted calibration trade-off hold irrespective of model size. Comparing ViT-B trained from scratch and from DINO, we see that calibration improvements are a consequence of pretraining, not of model size.

## A.10    Additional figures for ensembles

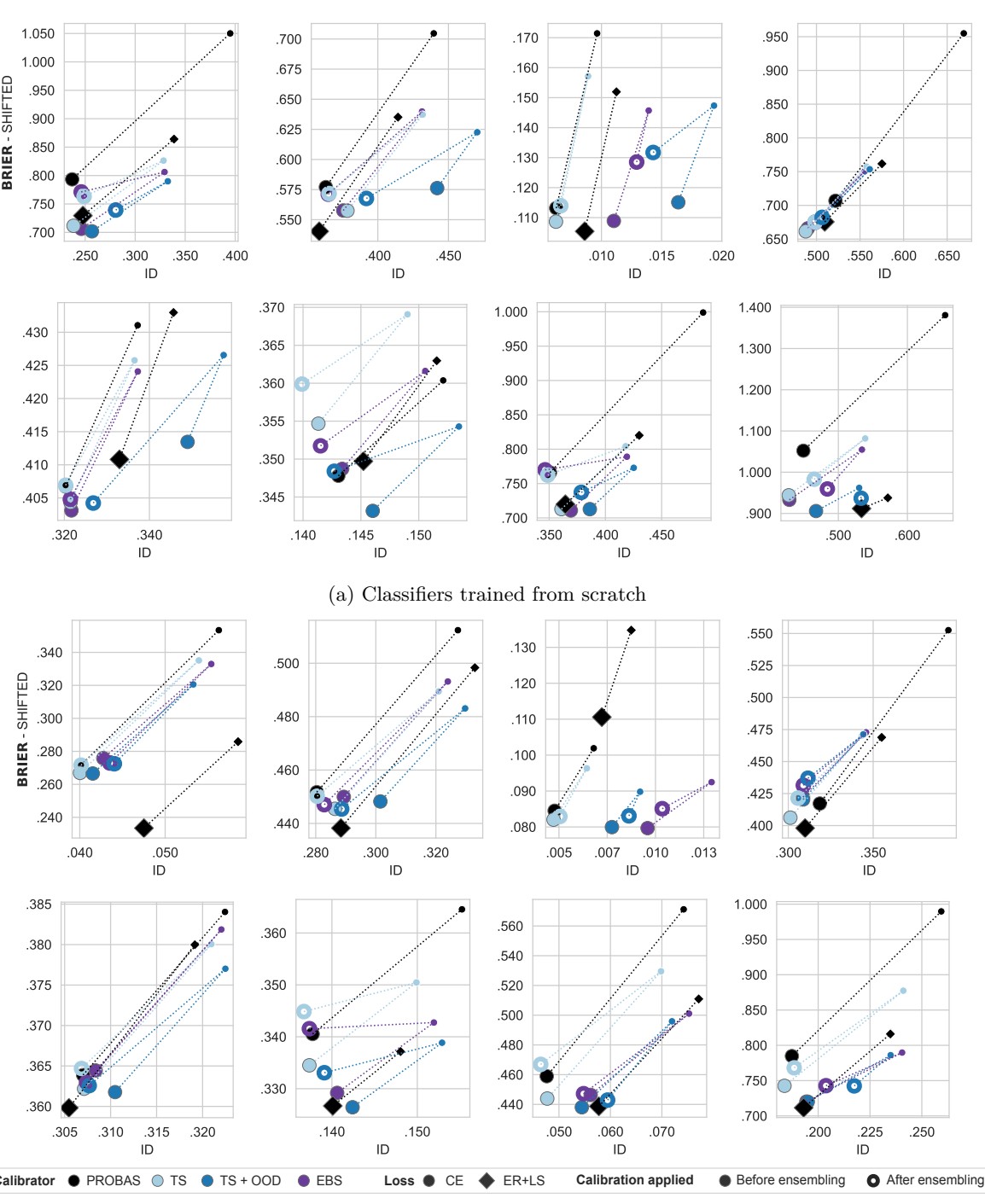

(a) Classifiers trained from scratch

(b) Classifiers finetuned from foundation models

Figure A.9: **Brier score of model ensembles, in function of post-hoc calibration method** (3 member-ensembles). Large dots denote calibration results for model ensembles, small dots the average calibration of individual ensemble members for reference. We compare the effect of (i) applying post-hoc calibration to ensemble members (before ensembling), (ii) applying post-hoc calibration after ensembling predictions.

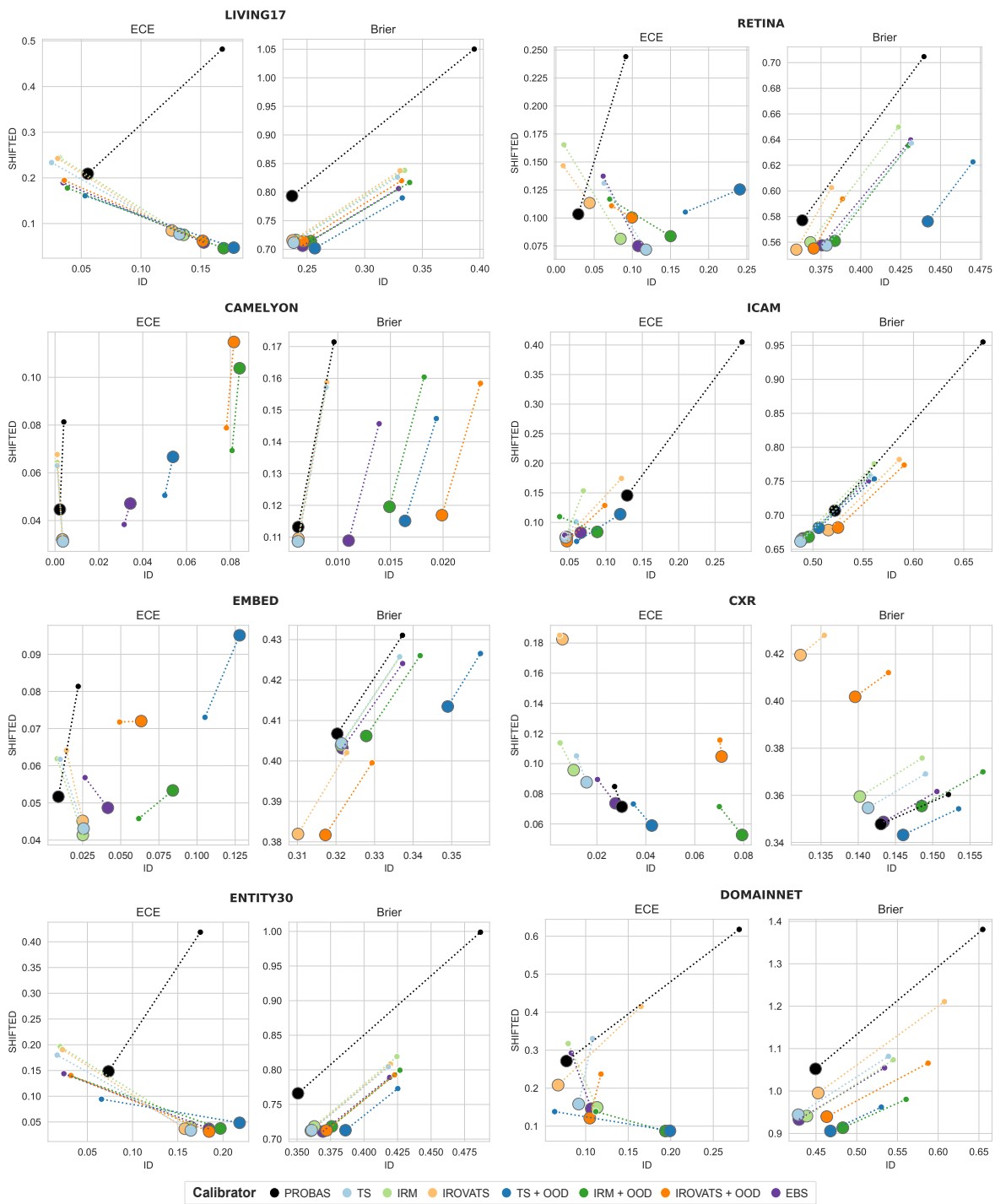

Figure A.10: **Ablation study of the effect of additional post-hoc calibrators of ensemble members** (prior to ensembling), for ensembles whose members were trained with CE (3 member-ensembles). Large dots denote calibration results for model ensembles, small dots denote the average calibration of individual ensemble members for reference.

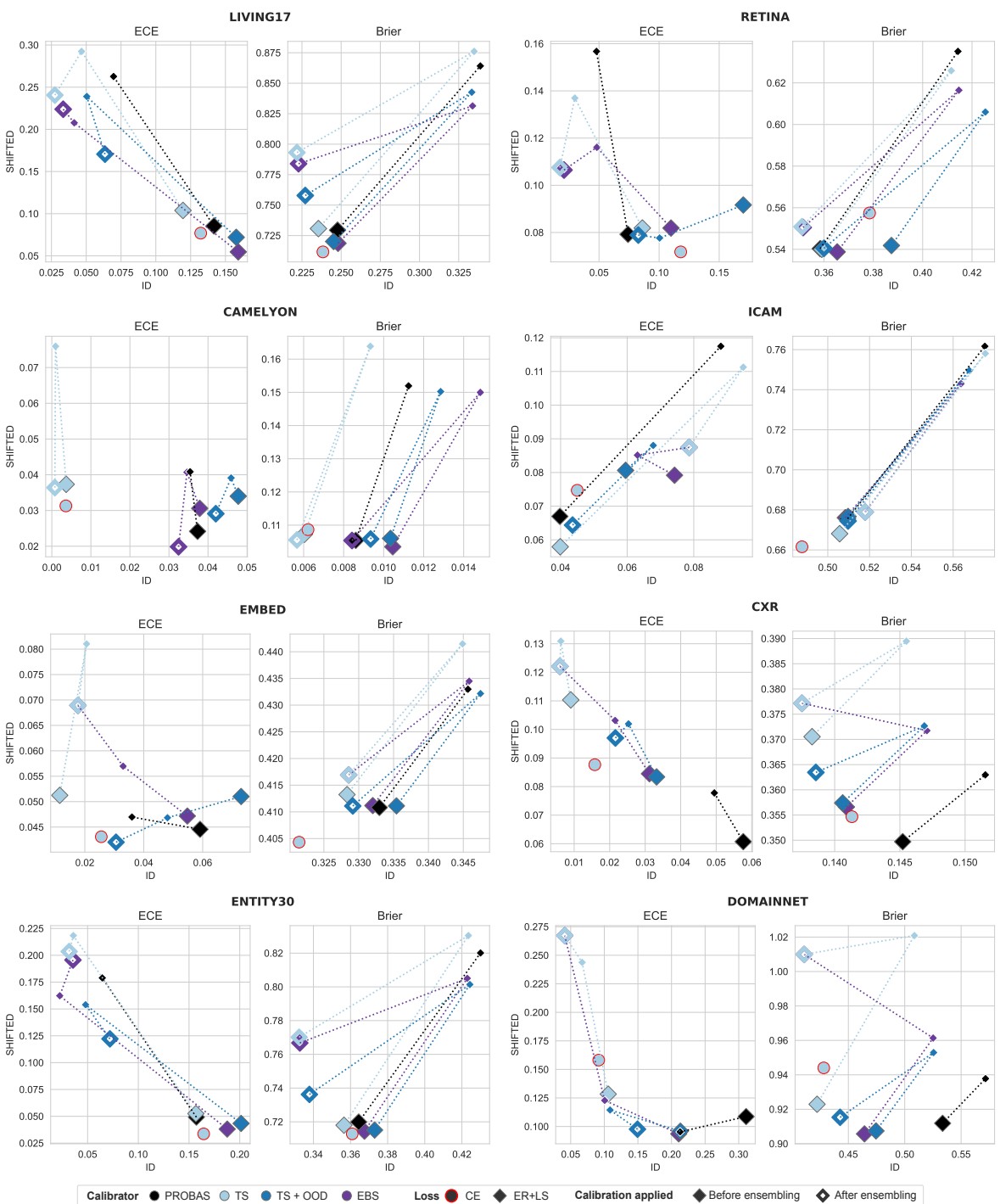

Figure A.11: **Calibration of model ensembles, in function of member calibration method, for ensembles whose members were trained with ERLS**. Large dots denote calibration results for model ensembles, small dots denote the average calibration of individual ensemble members for reference. We compare the effect of: (i) applying post-hoc calibration to ensemble members (prior to ensembling); (ii) applying post-hoc calibration after ensembling model predictions. We report calibration results for models trained with CE and with TS calibration post-ensembling as a baseline.

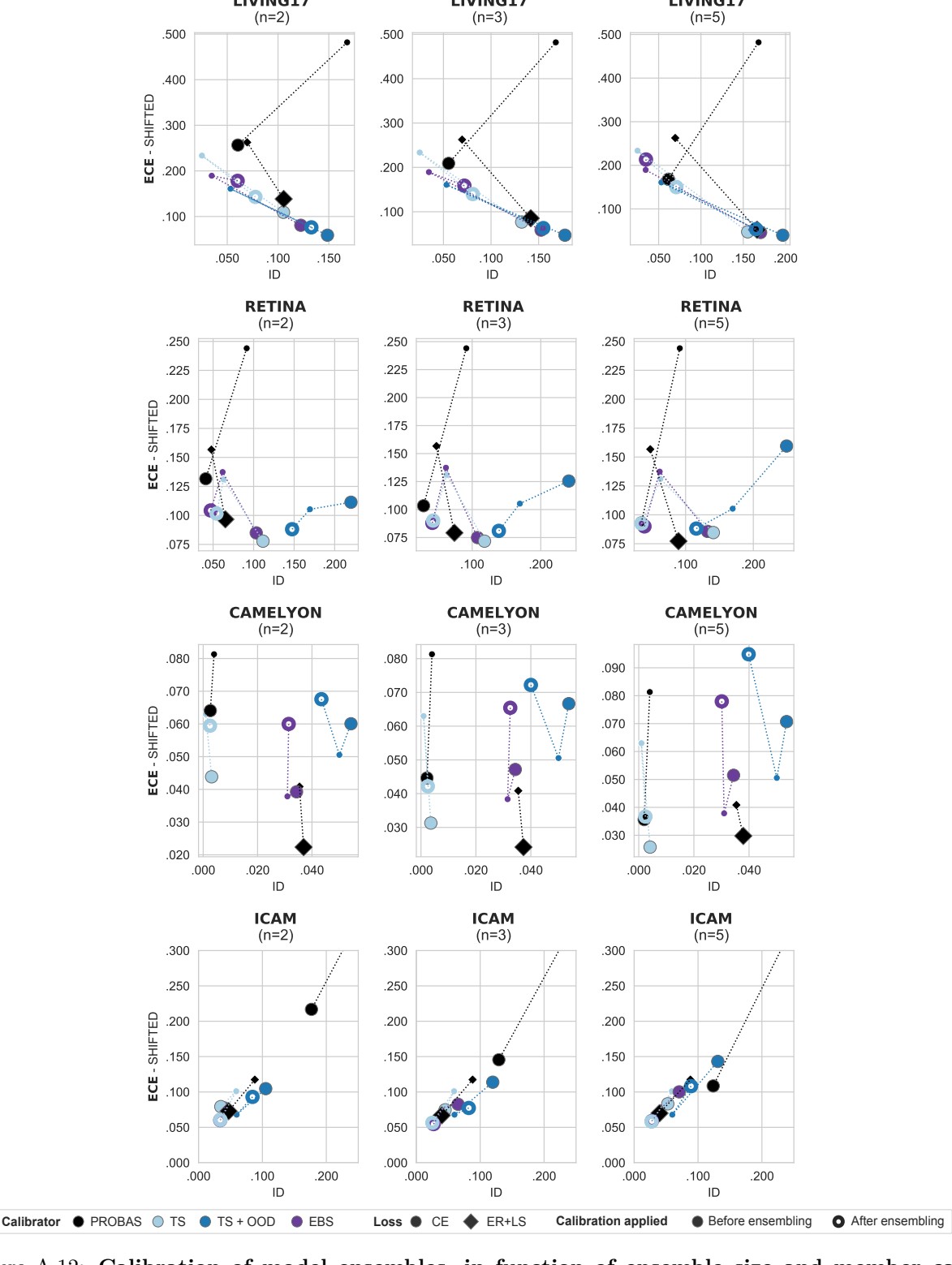

Figure A.12: **Calibration of model ensembles, in function of ensemble size and member calibration method, measured by ECE (1/2)**. Large dots denote calibration results for model ensembles, small dots denote the average calibration of individual ensemble members for reference. We compare the effect of: (i) applying post-hoc calibration to ensemble members (prior to ensembling); (ii) applying post-hoc calibration after ensembling model predictions. We report calibration results for models trained with CE and with TS calibration post-ensembling as a baseline.

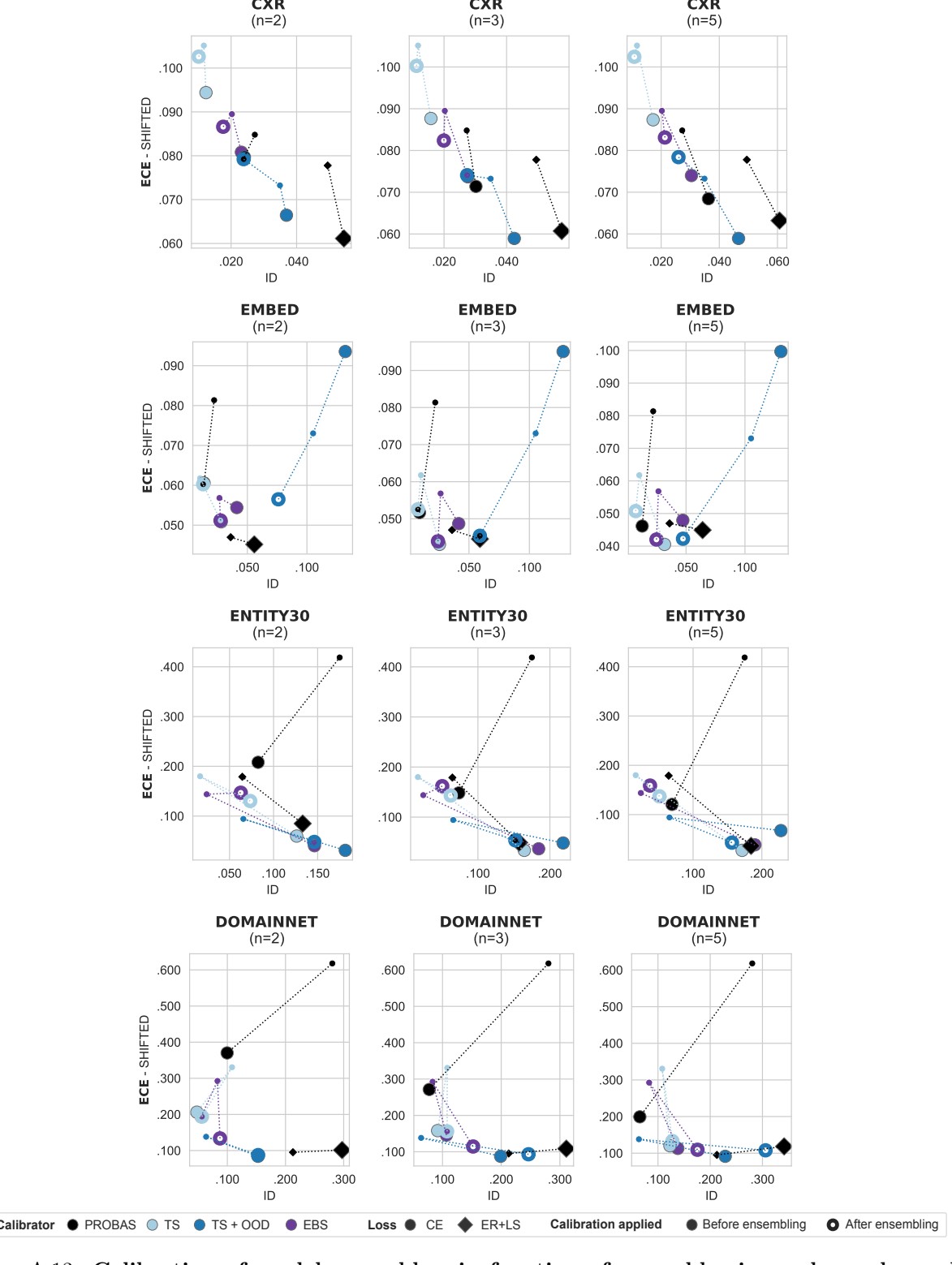

Figure A.13: **Calibration of model ensembles, in function of ensemble size and member calibration method, measured by ECE (2/2)**. Large dots denote calibration results for model ensembles, small dots denote the average calibration of individual ensemble members for reference. We compare the effect of: (i) applying post-hoc calibration to ensemble members (prior to ensembling); (ii) applying post-hoc calibration after ensembling model predictions. We report calibration results for models trained with CE and with TS calibration post-ensembling as a baseline.

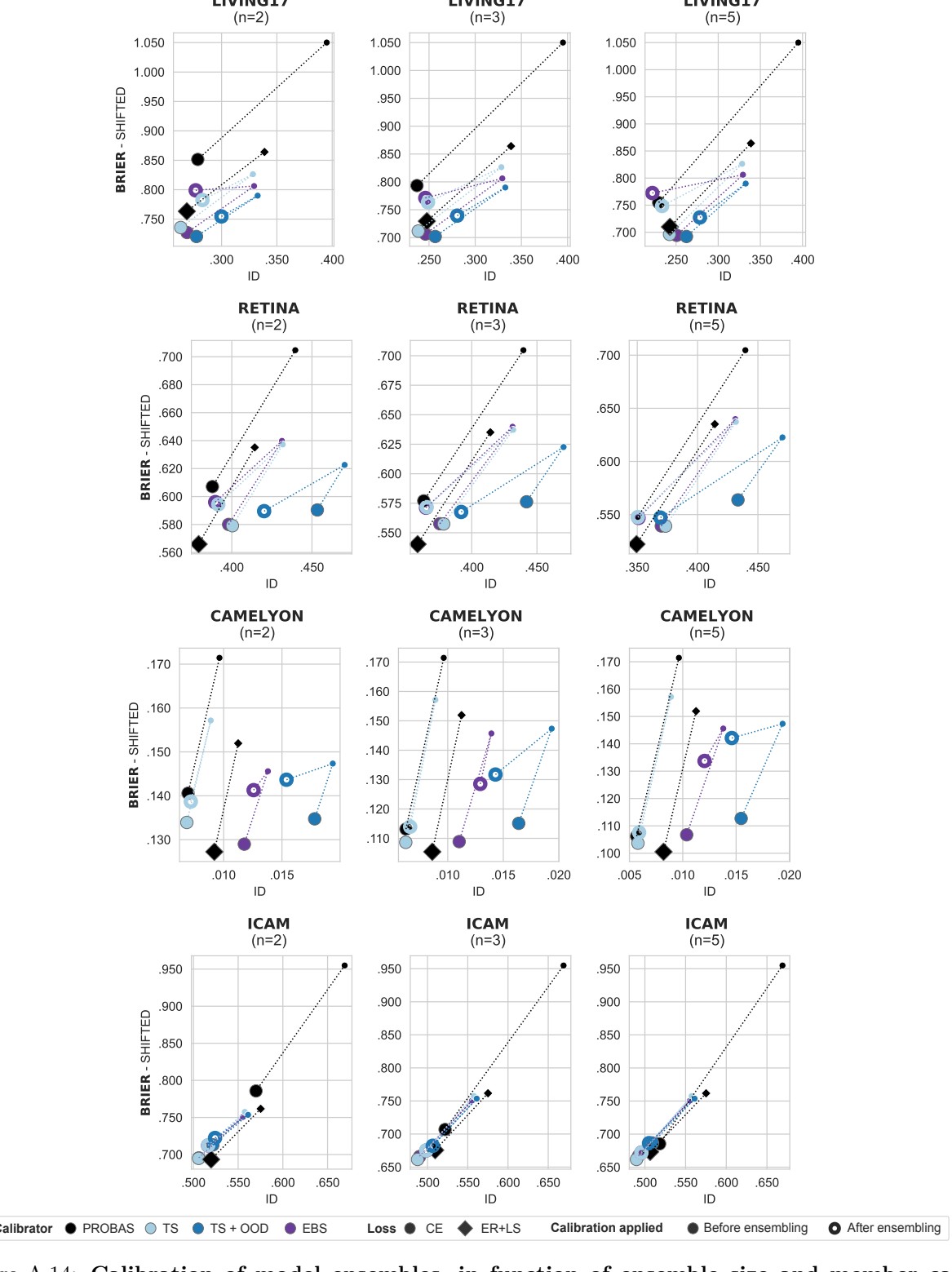

Figure A.14: **Calibration of model ensembles, in function of ensemble size and member calibration method, measured by Brier Score (1/2)**. Large dots denote calibration results for model ensembles, small dots denote the average calibration of individual ensemble members for reference. We compare the effect of: (i) applying post-hoc calibration to ensemble members (prior to ensembling); (ii) applying post-hoc calibration after ensembling model predictions. We report calibration results for models trained with CE and with TS calibration post-ensembling as a baseline.

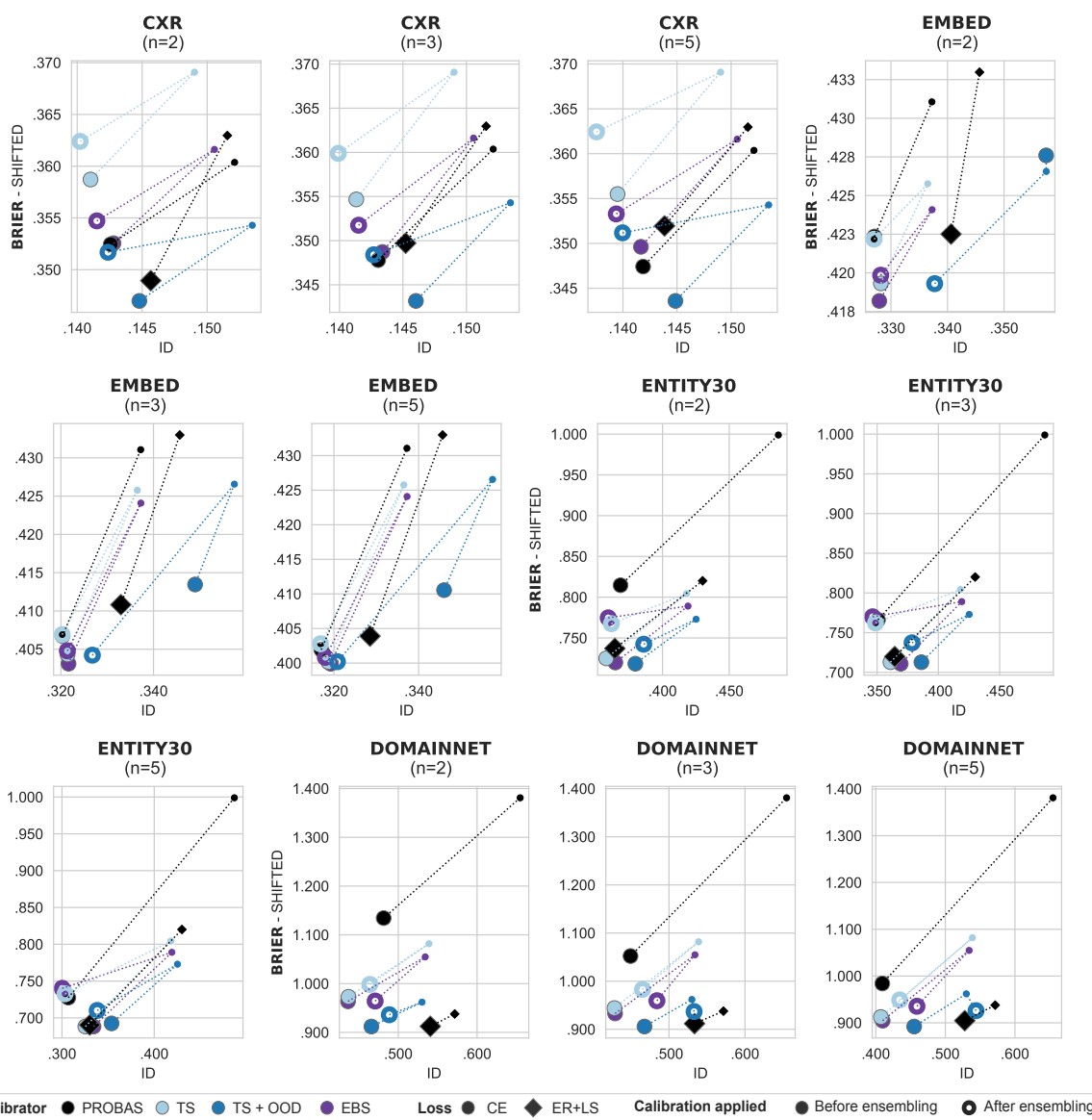

Figure A.15: **Calibration of model ensembles, in function of ensemble size and member calibration method, measured by Brier Score (2/2)**. Large dots denote calibration results for model ensembles, small dots denote the average calibration of individual ensemble members for reference. We compare the effect of: (i) applying post-hoc calibration to ensemble members (prior to ensembling); (ii) applying post-hoc calibration after ensembling model predictions. We report calibration results for models trained with CE and with TS calibration post-ensembling as a baseline.

