# OpenReview forum: "Where are we with calibration under dataset shift in image classification?"
_TMLR — Accepted by TMLR_

### Review · Reviewer_NV3v · 2025-08-07

**Summary Of Contributions:**

This paper conducts extensive experiment on existing calibration methods under real-world dataset shift. The author uses various image recognition dataset with natural dataset shift and evaluates several in-training and post-hoc calibration methods on random initialization and on foundation model. The author then describes the practical guideline about the calibration method to be used.

**Audience:**

Yes

**Audience Explanation:**

The contribution is the extensive study of existing calibration methods in realistic setting. The results would be interesting for the researchers in this area.

**Broader Impact Concerns:**

I see no broader impact section, but since the contribution is the experimental evaluation of existing methods on existing datasets, the risk of ethical problem is small.

**Claims And Evidence:**

Yes

**Claims Explanation:**

The author evaluates multiple methods on several real-world image recognition benchmarks to obtain the guideline. Therefore, I think the result is reliable.

One of my concerns is the number of trials. If each setting is evaluated once, it would be better to conduct evaluation multiple times and plot the standard deviation to ensure more reliability.

**Requested Changes:**

It would be clearer to visualize the example dataset images corresponding to Table A.1. so that we can see the dataset shift easily.

Upon acceptance, it would be better to publish the trained model in the experiment so that the reader can reproduce the results easily.

---

### Review · Reviewer_kehV · 2025-08-20

**Summary Of Contributions:**

The paper conducts a large-scale experimental evaluation on model calibration under real-world dataset shifts in image classification. The authors benchmark 5 in-training calibration strategies (e.g., label smoothing, entropy regularization) and 10 post-hoc calibration methods (e.g., temperature scaling, EBS) across 8 diverse datasets (natural and medical imaging) with realistic shifts (e.g., scanner changes, population shifts). Based on some interesting findings, the authors provide practical guidelines for constructing models with robust calibration under such shifts.

**Audience:**

Yes

**Audience Explanation:**

Model calibration is a meaningful and trustworthy topic in many fields. This paper focuses on calibration under dataset shift in image classification, providing valuable findings and inspiring insights for many TMLR's audience.

**Claims And Evidence:**

Yes

**Claims Explanation:**

## Strengths
1.  The authors conduct a comprehensive evaluation with breadth and diversity. This study covers 420 models from both random and foundation model initializations across 8 datasets with several in-training and post-hoc calibration techniques.

2. This paper is easy to read and uses many tables and figures for an intuitive visualization, e.g., Figure 1 shows an overview of the calibration study.

3. The paper has a clear structure and enough workload. The experiments are extensive and include various models with different calibration strategies.

**Requested Changes:**

This paper is very comprehensive and solid, so I recommend it for acceptance with a minor revision.
Regarding the model ensembles, the ensemble size is fixed as 3 in the experiments; scalability with size (k=1…9) is unstudied, leaving unanswered whether the reported calibration gains plateau or continue to improve as more models are added. It could be better to include a figure showing the calibration varies with the ensemble size k ∈ {1,3,5,9}.

---

### Review · Reviewer_5gJ9 · 2025-08-24

**Summary Of Contributions:**

The paper presents a comprehensive benchmark of methods for calibrating deep neural networks under dataset shift in image classification. Motivated by an overview of related works, the benchmark is designed to span diverse combinations of natural dataset shifts, randomly initialized and pretrained architectures, in-training and post-hoc calibration methods, as well as single models and ensembles. Performance is evaluated using the expected calibration error and the Brier score, supported by statistical tests. Based on the results, the authors analyze several (often underexplored) factors influencing calibration quality, highlighting the substantial benefits of fine-tuning foundation models over training from scratch and showing that post-hoc calibration largely eliminates the need for in-training calibration. The key findings are distilled into a straightforward pipeline that guides practitioners in selecting the most effective combination of calibration method, architecture, and training procedure, tailored to their computational budget and the availability of semantic out-of-distribution data. Moreover, the provided codebase delivers a good starting point for adopting such combinations toward specific learning tasks.

**Additional Comments:**

- In Figure 6, some numbers on the $x$-axis overlap.
- In scientific writing, “do not” is preferred over “don’t.”
- Why is EBS presented prominently as Algorithm 1 in the appendix?
- In Figure A.1, the top of the dataset name “RETINA” is cut off.
- The first paragraph of Section 4.1.1 ends with two periods.
- In Figure 4, should the $y$-axis label be “shifted” rather than “OOD”?
- The references show inconsistencies, for example, in how NeurIPS papers are cited.
- Do you mean ER+LS instead of ERLS in the caption of Figure A.9 and Appendix A.5?

**Audience:**

Yes

**Audience Explanation:**

- Calibration is a highly relevant and active research topic, especially in the presence of dataset shifts.
- By considering both modern and established calibration methods and architectures, the paper offers conclusions that are valuable to the broader research community and practitioners.
- The provided codebase is extensible and can support future studies, including the development of new calibration methods.
- The shift-aware testbed may also benefit adjacent research areas, such as active learning under distribution shifts.

**Broader Impact Concerns:**

The paper lacks a dedicated discussion of broader impacts. The authors could briefly address the importance of well-calibrated probabilities as a requirement in many real-world applications and explain how their practical guide can support such use cases.

**Claims And Evidence:**

Yes

**Claims Explanation:**

- The takeaways are well grounded in extensive benchmark results spanning numerous calibration pipelines and statistical tests.
- Most calibration methods considered, such as label smoothing, temperature scaling, and entropy regularization, are well established in practice, which increases the practical value of the proposed decision pipeline.
- The overview of the related work identifies underexplored aspects and, thus, demonstrates the need for the presented benchmark.
- The dataset collections vary in class counts, domains, and shift types, providing a realistic and challenging testbed for evaluating calibration methods on both in-distribution and shifted data.
- The code is publicly available and generally well documented, enabling researchers to reproduce results and practitioners to apply the calibration pipelines in their own settings.
- The paper is clearly structured, with each takeaway explicitly tied to specific sections, and the graphical presentation of results is intuitive, making the authors’ claims easy to evaluate.

**Requested Changes:**

- While the authors provide a solid overview of prior work on calibration, including under dataset shift, a tabular comparison highlighting how this benchmark differs from earlier evaluations would better justify its necessity. For instance, the authors could contrast their number of calibration methods, datasets, and architectures against those considered in earlier studies.
- The paper does not describe the training procedures and optimizer hyperparameters, which can only be inferred from the codebase. This is particularly relevant for fine-tuning foundation models, where choices such as full fine-tuning with block-wise learning rate decay versus fine-tuning only selected blocks can significantly affect results.
- The exclusive focus on image classification is restrictive, since other modalities also require calibrated neural networks. A more detailed rationale for this scope limitation would strengthen the paper.
- Although the Nemenyi test is still widely used, it has well-known drawbacks, such as the dependence of pairwise outcomes on which additional methods are included in the comparison. Benavoli et al. [1] recommend pairwise Wilcoxon signed-rank tests with proper correction for family-wise error. The authors should either explain their choice of test procedure or adopt more up-to-date approaches.
- The paper would benefit from a dedicated section on current limitations and future research directions, such as extending the analysis to other modalities or integrating hyperparameter optimization.

**Reference:**
- [1] Benavoli, Alessio, Giorgio Corani, and Francesca Mangili. "Should we really use post-hoc tests based on mean-ranks?." The Journal of Machine Learning Research 17, no. 1 (2016): 152-161.

---

### Review · Reviewer_4DCR · 2025-08-24

**Summary Of Contributions:**

This work provides extensive evaluation studies on post-hoc calibration methods and their interactions with some in-training calibration strategies (e.g., label smoothing). It focuses on the natural shifts in real-world image classification tasks. The findings and observations are clearly conveyed.

**Additional Comments:**

In the experiments, the authors consider the pre-trained models. My additional question is do the authors have the insights will the drawn conclusions are robust to the type of backbones?

**Audience:**

Yes

**Audience Explanation:**

The work focuses on calibration techniques in real-world image classification under natural distribution shifts. The findings can contribute to a broader context of robustness and trustworthiness of AI.

**Broader Impact Concerns:**

Not applicable from my side.

**Claims And Evidence:**

Yes

**Claims Explanation:**

The evaluation motivation, pipelines are clearly described.

Most of the evaluation details are clearly stated.

The implementation codes are attached, which is a plus.

In terms of the presentation, it is well structured, easy to follow.

In terms of the conveyed findings from the evaluation, the experiment can support their claims.

**Requested Changes:**

For me, the work only requires some minor changes:

1. Consider adding a table to highlight the differences compared to the existing studies.

2. Consider adding some examples to visualize some details of the natural distribution shifts in the applied datasets. Because synthetic distribution shifts (corrections) are quite easy to picture for readers.

3. Consider increasing the size of the main result figures for better illustration.

---

### Author Response · Authors · 2025-09-02
**Point-by-point response to reviewers**

We thank all reviewers for taking the time to review our manuscript.

We are pleased to see that all reviewers recognised the value of the manuscript for the TMLR community, and all agree that our claims are ‘supported by accurate, convincing and clear evidence’.

In the following, we detail the changes made to the manuscript in response to reviewers’ requests. All changes appear **in blue** in the updated version of the manuscript:

* [4DCR, 5gJ9] **Adding a table to highlight the differences compared to the existing studies.**
We thank reviewers 4DCR, 5gJ9 for their suggestions. We have added such a comparative table in **Table A.2.**, referenced in the main paper at the end of section 2.4.
* [4DCR, NV3v] **Adding some examples to visualize details of the natural distribution shifts in the applied datasets.**
We thank the reviewers 4DCR, NV3v for suggesting to add some visual examples of the various shifts studied in this paper. We have added the requested figure to the appendix, please see **Fig A.1 & Fig A.2**.
* [kehV] **Effect of ensemble size on reported conclusions.**
We thank the reviewer for their question. For completeness, we have now added an additional ablation study on the size of the ensemble, varying ensemble size n=2,3,5, please refer to newly added **Fig A.12 to A.15**. This experiment shows that  _“calibration gains are readily visible with ensembles with only two members only. Gains on shifted calibration increase slightly as the number of members increases, however we observe a plateau-effect with only minimal improvements gained from 3 to 5 ensemble members.”_ Moreover in terms of post-hoc calibration effect, the ablation study also shows that our conclusions hold irrespective of model size. We have added these additional insights to the main text in **sections 4.3.1 and 4.3.2**.
* [5gJ9] **Details on hyperparameters.**
We have added the requested details (optimiser, loss, learning rate) in Appendix **A.4**, and added a reference to this new section in the main paper in section 3.2.
* [5gJ9] **Adding a paragraph on limitations of the scope of the study in the conclusion.**
We have added the following new paragraph at the end of the **conclusion**, referencing complementary studies for other tasks (regression, segmentation), as well as emphasising limitations of the current study: _‘In this study, we focused exclusively on robustness of calibration under dataset shifts for image classification. It is worth noting that building reliable and well-calibrated uncertainty estimates, robust across dataset shifts, is equally relevant for other tasks. \cite{gustafsson_how_2023} for example studied the reliability of existing uncertainty estimates in the context of regression tasks, whereas \cite{jorge_reliability_2023} similarly studied robustness and calibration of uncertainty estimates in segmentation. Moreover, the guidelines proposed here, are based on our extensive evaluation of currently commonly used post-hoc and in-training calibration methods, and these may require updating as new calibration methods appear in the future. To this end, our publicly available benchmarking codebase provides a comprehensive evaluation framework and facilitates future benchmarking efforts for calibration under shifts in image classification.’_
* [5gJ9] **Use of the Nemenyi test for statistical analysis in the appendix.**
We used the Nemenyi test as it is a widely recognised statistical test when comparing more than two groups (after a significant Friedman test).  See for example lecture notes from  https://minkull.github.io/slidesCISE/08-evaluating-comparing-II.pdf, blog post from https://lab.rivas.ai/?p=2665
* [5gJ9] **Minor typos.**
We thank the reviewer for highlighting these typos, we have corrected all of them in the new version of the manuscript.

---

> ### Comment · Reviewer_5gJ9 · 2025-09-21
>
> Dear authors,
>
> Many thanks for your concise response, including the proper changes, further improving your manuscript and its contributions. After rechecking these changes and rereading some other parts of your paper, I suggest the following **(very) minor changes**:
> - Before introducing the ECE in Eq. (1), you define $\mathrm{Conf}(B_m) = \sum_{\hat{c}^i \in B_m} \hat{c}^i$ as average confidence and $\mathrm{Acc}(B_m) = \sum_{\hat{c}^i \in B_m} \mathbb{I}[y^i = \hat{y}^i]$ as accuracy for the $m$-th bin. I assume both terms miss the normalization by $|B_m|$ as the number of samples in the $m$-th bin. For convenience, I also would propose to use the sample index $i \in B_m$ as the iteration variable for both sums.
> - In Fig. 1, you could introduce most of the abbreviations, in particular for the in-training and post-hoc calibration methods, in the boxes, which would further facilitate the reading of the guidelines in Fig. 2.
> - In Table A.2, it should be "Type of **s**hifts" instead of "Type of **S**hifts".
> - Recheck whether all references are up to date. For example, you reference "How Reliable is Your Regression Model's Uncertainty Under Real-World Distribution Shifts?" as an arXiv paper despite being published with TMLR.
> - In Appendix A.5.2, add missing math mode to write "... where $C$ is the number of classes ..." instead of " ... where C is the number of classes ...",
> - In Algorithm 2, remove the dot at the end of the comment "... here TS.".
> - Make your mathematical notation more consistent. For example, you write "classifier $f(\mathbf{x})$" but then denote the input samples as $x$ in Algorithm 2. Additionally, you iterate over the sample indices $i \in \mathcal{D}$ although $\mathcal{D}$ consists of sample-label pairs in Algorithm 1. Check also the assignment ($=$ vs. $\leftarrow$) of variables for both algorithms.

---

### Decision · Action_Editor_VS61 · 2025-10-04

**Recommendation:** Accept as is

**Audience:**

Yes

**Audience Explanation:**

Calibration and dataset shift are important topics of machine learning.

**Claims And Evidence:**

Yes

**Claims Explanation:**

The claims are supported by accurate and convincing evidence from a large-scale empirical study. The conclusions are directly derived from rigorous experiments across eight real-world datasets, multiple model architectures, and extensive benchmarking of calibration methods.